# CD74 supports accumulation and function of regulatory T cells in tumors

Elisa Bonnin [1,2], Maria Rodrigo Riestra[1,2], Federico Marziali[1,2], Rafael Mena Osuna[1,2], Jordan Denizeau[1,2], Mathieu Maurin[1], Juan Jose Saez [1], Mabel Jouve[1], Pierre-Emmanuel Bonté [1], Wilfrid Richer[1,2], Fabien Nevo[3], Sebastien Lemoine[3], Nicolas Girard[1,4,5], Marine Lefevre[6], Edith Borcoman[7], Anne Vincent-Salomon [5,8], Sylvain Baulande [9], Helene D. Moreau[1], Christine Sedlik[1,2], Claire Hivroz [1], Ana-Maria Lennon-Duménil[1], Jimena Tosello Boari [1,2,10] ✉ & Eliane Piaggio [1,2,3,10] ✉

Regulatory T cells (Tregs) are plastic cells playing a pivotal role in the maintenance of immune homeostasis. Tregs actively adapt to the microenvironment where they reside; as a consequence, their molecular and functional profiles differ among tissues and pathologies. In tumors, the features acquired by Tregs remains poorly characterized. Here, we observe that human tumor-infiltrating Tregs selectively overexpress CD74, the MHC class II invariant chain. CD74 has been previously described as a regulator of antigen-presenting cell biology, however its function in Tregs remains unknown. CD74 genetic deletion in human primary Tregs reveals that CD74KO Tregs exhibit major defects in the organization of their actin cytoskeleton and intracellular organelles. Additionally, intratumoral CD74KO Tregs show a decreased activation, a drop in Foxp3 expression, a low accumulation in the tumor, and consistently, they are associated with accelerated tumor rejection in preclinical models in female mice. These observations are unique to tumor conditions as, at steady state, CD74KO-Treg phenotype, survival, and suppressive capacity are unaffected in vitro and in vivo. CD74 therefore emerges as a specific regulator of tumor-infiltrating Tregs and as a target to interfere with Treg anti-tumor activity.

Regulatory T cells (Tregs) constitute a heterogeneous population essential to maintain homeostasis and to control immune responses. However, Tregs also inhibit anti-tumor immunity. Treg plasticity enables them adapting to the microenvironment in which they reside by developing optimized mechanisms to efficiently regulate immune responses according to the tissue. Nevertheless, Treg adaptation to

external cues can also induce Treg instability, which is associated to a switch from suppressive to pro-inflammatory functions, and can lead to the development of autoimmunity, or alternatively to reinforce anti-tumor responses. Thus, thorough characterization of Treg features associated to individual tissues or pathologies should help in the design of optimized immune therapies. For cancer application, an

[1]INSERM U932 Immunity and Cancer, PSL University, Institut Curie Research Center, Paris, France. [2]Department of Translational Research, PSL Research University, Institut Curie Research Center, Paris, France. [3]Egle Therapeutics, Paris, France. [4]Paris Saclay University, UVSQ, Versailles, France. [5]Institut du Thorax Curie Montsouris, Institut Curie, Paris, France. [6]Pathology Department, Institut Mutualiste Montsouris, Paris, France. [7]Department of Drug Development and Innovation (D3i), Institut Curie, Paris, France. [8]Diagnostic and Theranostic Medicine Division, Institut Curie, PSL Research University, Paris, France. [9]Institut Curie Genomics of Excellence (ICGex) Platform, PSL Research University, Institut Curie Research Center, Paris, France. [10]These authors jointly supervised this work: Jimena Tosello Boari, Eliane Piaggio. ✉e-mail: jimena.tosello@curie.fr; eliane.piaggio@curie.fr

optimal strategy would be to inhibit Treg function only in the tumor, while sparing Tregs in the rest of the body. Along these lines, although some tumor-associated Tregs (tumTregs) molecular characteristics have been described, the identification and characterization of tumTreg unique biomarkers remain understudied.

Using available transcriptomic data from single-cell RNA and bulk RNA sequencing from CD4+ T cells from cancer patients, we identified the cluster of differentiation 74 (CD74) to be a specific biomarker of tumTregs (manuscript under review). CD74 was found to be overexpressed by tumTregs compared to matched blood Tregs; and to tumor and blood conventional CD4+ T cells (Tconvs). CD74 was initially described as the invariant chain of the major histocompatibility complex class-II (MHC-II), and therefore its expression was thought to be mainly restricted to antigen-presenting cells (APCs)[1]. This type II transmembrane protein binds to MHC class II into the endoplasmic reticulum and follows the secretory route to reach the plasma membrane. There, CD74 is rapidly endocytosed in an AP-2-dependent manner, which limits its residency at the cell surface and promotes its accumulation within the endolysosomal compartment[2,3]. Additional functions have been described for CD74 in the regulation of NFκB-dependent gene transcription[4] and of the migratory properties of APCs[5]. Two non-exclusive mechanisms could account for the later: (1) regulation of the actomyosin cytoskeleton through its interaction with the protein motor myosin II, and (2) regulation of chemotaxis through its interaction with the macrophage-migration inhibitory factor (MIF), for which CD74 was shown to act as a surface receptor. Although Tregs were described to upregulate MHC-II-related molecules during inflammation[6–8] or upon activation[9], whether CD74 is stably expressed in these cells and how it impacts on their behavior and function remains largely unknown.

To address the role of CD74 in Tregs, we generate CRISPR-Cas9 CD74 knockout (CD74KO) cells from primary human Tregs. Here we show that CD74KO Tregs are exclusively dysfunctional in the tumor tissue, where they display a defective capacity to maintain and to suppress. At odds, their suppressive activity remains unaffected in vitro, and in vivo, in a GVHD model. These results suggest that CD74 might play a specific role in tumTregs and identify this protein as a target to selectively manipulate Treg migration and suppressive function in cancer.

## Results

### Tumor-associated Tregs upregulate surface CD74
To evaluate whether Tregs present in the tumor microenvironment (TME) exhibit specific molecular features that distinguish them from Tregs in other tissues, we used an in silico approach. We analyzed a single-cell RNA sequencing (scRNAseq) dataset of CD4+ T conventional (Tconv) and Treg cells from non-small cell lung cancer (NSCLC) patients. Among the genes with a significantly higher differential expression in tumor-Tregs (tumTregs) compared to tumor-Tconvs, we observed several ones associated to activated tumTregs like CD25, HLA-DR, CTLA-4 and CCR8, and also CD74, whose biological function in Tregs has not been previously described (Fig. 1A). We extended this analysis to blood-derived Tregs and Tconvs from the same scRNAseq dataset (Figs. 1B and S1A) and observed that tumTregs overexpressed CD74 transcripts as compared to blood Tregs and to blood and tumor Tconvs. These data were confirmed in published bulk RNAseq transcriptomic data of sorted CD4+ T cells from NSCLC (Fig. 1C) and other cancer types (Fig. S1B).

Next, we assessed CD74 protein expression level at the surface of Tregs (CD4+Foxp3+) and Tconvs (CD4+Foxp3-) from paired blood and tumor samples of NSCLC patients by flow cytometry (Fig. S1C). Very few blood and tumor Tconvs, as well as blood Tregs express CD74 at the surface, while 6 to 22% tumTregs express it, depending on the patients (Fig. 1D, E). Thus, as pointed out by the transcriptomic data, CD74 appears as a specific biomarker of tumTregs. Moreover, in CD8+

T cells from the tumor, CD74 surface expression is almost undetectable (Figs. 1D, E and S1C, S1D). Similar results were observed in breast cancer samples, suggesting that expression of CD74 in tumTregs could be shared across different cancer types (Fig. S1C). Furthermore, Tregs present in the metastatic lymph nodes presented an intermediate expression of CD74 at the surface (Fig. S1D). Additionally, a similar pattern of significantly higher expression of surface CD74 on human Tregs in the tumor over CD8+ T cells and CD4+ Tconvs was observed in a humanized mouse model of *Nod Scid Gamma (*NSG) mice engrafted with human tumor and T cells (Fig. S1E).

To investigate whether epigenetic imprinting underlies the differential expression of CD74 in Tregs versus Tconvs, we analyzed single-cell ATAC sequencing data from NSCLC patient-derived Tregs and Tconvs (Tosello dataset). We mapped the accessible chromatin at the CD74 locus, which marks areas of active gene transcription. As shown in Fig.1F, Tregs and Tconvs differ in the open chromatin landscape surrounding the CD74 gene according to the tissue. While only one intronic peak is more accessible in blood Tregs compared to blood Tconvs (Pval ≤ 0.05 & Log2FC ≥ 0.5) (chr5:150409694–150410194); four peaks are more accessible in tumTregs compared to tumTconvs (chr5:150397090–150397590, chr5:150397621–150398121, chr5:150398582–150399082, chr5:150399367–150399867), all corresponding to intronic regions of CD74. Notably, in Tregs two of these opened peaks (chr5:150398582–150399082, chr5:150399367–150399867) correlate with CD74 gene expression, and are predicted to bind several transcription factors, including notably FOXP3, and C/EBPbeta, c-Jun, c-Ets-1, GR, and STAT4 among others[10–12]. Overall, these results suggest that distinct CD74 expression in Tregs and Tconvs is partly driven by epigenetic factors.

Using peripheral blood mononuclear cells (PBMCs) from healthy donors (HD), we observed that CD74 surface expression in Tconvs and Tregs was very low and similar to blood cells from NSCLC patients (Fig.1 G, H). This low detection of CD74 by surface staining is consistent with the elevated recycling property of this molecule[13,14]. In sharp contrast, 80.8 ± 4.3% (mean ± SD) of Tconvs and 99.6 ± 0.3% (mean ± SD) of Tregs expressed CD74 intracellularly. To assess whether CD74 surface expression could be induced by pro-inflammatory cytokines or by factors secreted by the tumor, we incubated human Tregs with IFNα, IFNγ, or tumor supernatant (tuSN); but no induction was observed (Fig. S1F).

Our results indicate that CD74 expression is selectively upregulated at the surface of tumTregs. They further suggest that CD74+ Tregs represent a distinct population of activated tumTregs and that CD74 could play a specific function in these cells.

### The survival, phenotype and suppressive capacity of Tregs are not affected by CD74 deficiency in vitro
To understand the biological role of CD74 on Tregs, we generated Tregs genetically KO for this gene using the clustered regularly interspaced short palindromic repeats (CRISPR)-associated protein 9 (CRISPR-Cas9) technology (Fig. 2A). Briefly, Tregs were FACS-sorted from HD PBMCs (Fig. S2A), in vitro expanded, activated, and electroporated with a CRISPR-Cas9 ribonucleoprotein (crRNP) containing a duplex of CD74-targeting RNAs (CD74KO Tregs), or with an empty crRNP (control, WT Tregs). The fact that Tregs showed almost 100% of intracellular expression of CD74 (Fig. 1G, H), allowed monitoring the efficacy of CD74 gene deletion using FACS. As illustrated in Fig. 2B, 70−90% of Tregs lost CD74 protein expression 4 days upon electroporation. Figure S2B shows the compilation of CD74 KO efficacy data for all Treg donors used.

We first assessed the effect of CD74 on Treg survival and proliferation. Similar in vitro cell expansion was observed for cultured WT and CD74KO Tregs along a 20-day observation period (Fig. 2C). Upon 7 days of culture, no significant difference was observed in the protein expression of GITR, CCR8, ICOS, OX40, CTLA-4, CD25, TIGIT, 4−1BB

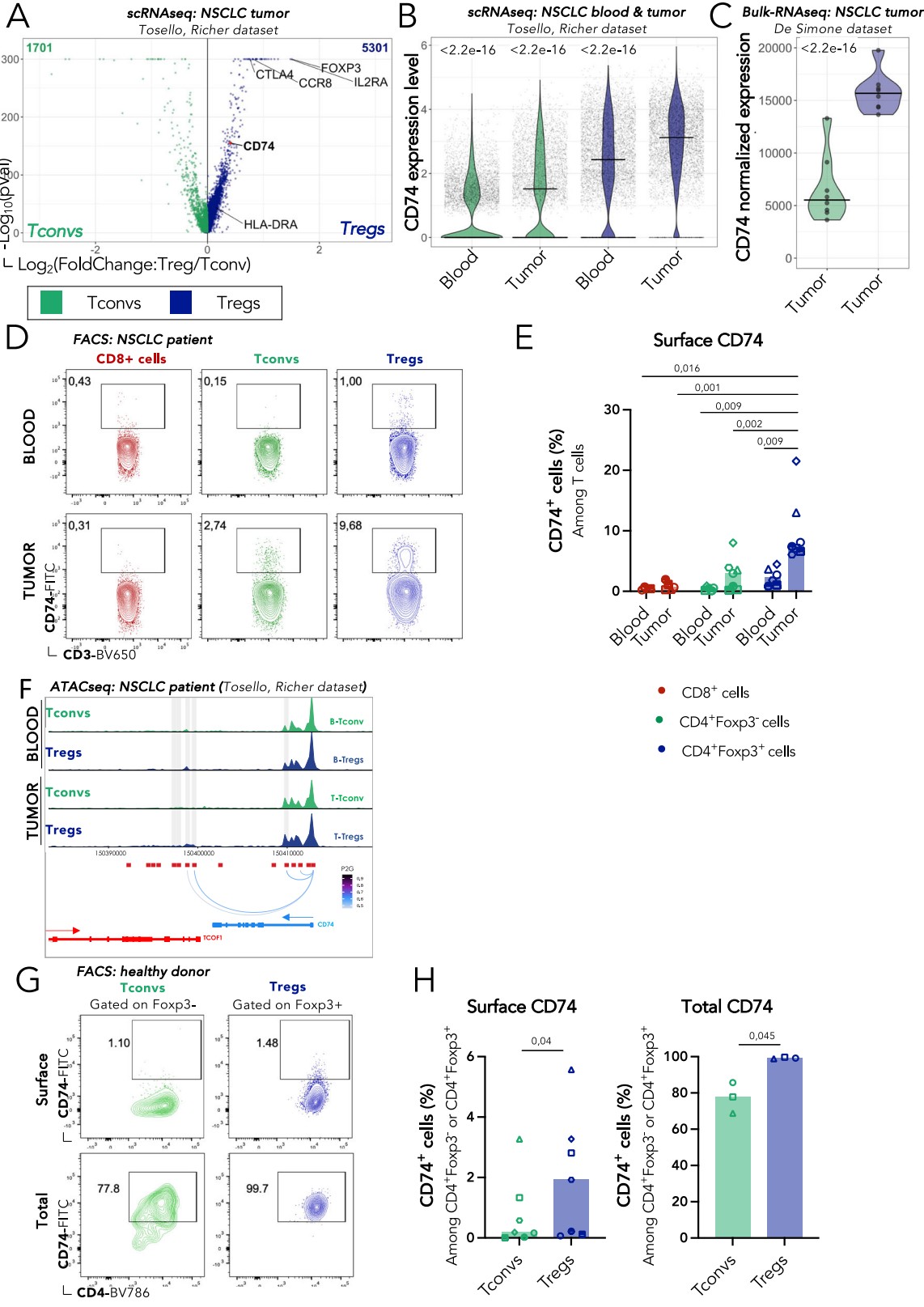

and Foxp3, while PD1 was mildly upregulated (Figs. 2D, E, S2C–E). As expected, expression of HLA-DR was statistically significantly reduced at the surface of CD74KO Tregs compared to their WT counterparts, consistent with CD74 being an MHC-II chaperone[15] (Figs. 2D–F and S2C, E). Finally, CD74 WT and KO Tregs exhibited a similar capacity to inhibit Tconv proliferation across the range of evaluated Treg:Tconv ratios (Fig. 2G).

Overall, these results indicate that CD74 is not essential for Treg proliferation, activation, and suppressive function in vitro.

### CD74-deficient Tregs conserve their suppressive function in a xeno-GvHD model

We next assessed the requirement of CD74 for Treg suppressive activity in vivo using a xeno-Graft versus Host disease (GvHD) model.

**Fig. 1 | CD74 is differentially expressed at the membrane of tumTregs compared with tumTconvs, peripheral Tregs and Tconvs. A, B** Single-cell RNA sequencing (scRNAseq) of CD4+ T cells from blood and tumor of 5 non-small cell lung cancer (NSCLC) patients (from: Tosello, Richer et al., under review, dataset available at EGAS50000000293). **A** Volcano plot (*P* value versus fold change) of the gene expression profile of Tregs and Tconvs. Genes highlighted in blue are overexpressed by tumTregs and highlighted in green by tumTconvs. **B** Violin plots of CD74 expression levels in Tregs and Tconvs from blood-tumor paired samples. **C** Violin plots of CD74 normalized expression from bulk RNA sequencing data (bulk-RNAseq) from sorted tumor CD4+ Tconvs and Tregs from NSCLC patients (from: De Simone et al. GEO access: GSE40419, design of 87 surgical samples[47]). **D, E** FACS analysis of CD74 expression in CD8+ cells (red), Tconvs (green), and Tregs (blue) from blood and tumor samples from NSCLC patients (*n* = 7). Representative dotplots (**D**) and percentages (**E**) of cells expressing surface CD74 (gating strategy in Fig. S1C). **F** Genome track showing peak accessibility of CD74 human gene loci and peak to genes links in Tconvs (green) and Tregs (blue) from single-cell ATACseq analysis of blood and tumor samples from two NSCLC patients (from: Tosello, Richer et al., under review, dataset available at EGAS50000000294). Surface and total FACS analysis of CD74 expression in Tregs and Tconvs from healthy donor (HD) PBMCs. Representative dotplots (**G**) and percentages (**H**) of surface (left, *n* = 7) and total (right, *n* = 3) expression of CD74 in Tregs (blue) and Tconvs (green). Statistical analyses are performed using unpaired *t*-test for tumTregs (**B, C**) or paired *t*-test between donor (**E, H**). *p* values are shown on the graphs. Source data are provided as a Source Data file.

In this model, GvHD is induced by the injection of human PBMCs into NSG immunodeficient mice, and the co-injection of Tregs limit the activation of the effector T cells, alleviating the clinical signs of the disease[16]. As illustrated in Fig. 3A, NSG mice were injected with PBMCs alone (control) (prepared from HLA.A2+ donors), or together with WT Tregs or CD74KO Tregs (prepared from HLA.A2- donors), which we named "expanded Tregs" (expTregs) to differentiate them from the Tregs initially present in the PBMC inoculum (Fig. S3A). We followed the kinetics of the injected expTregs in the blood (Fig. 3B). Both WT and CD74KO Tregs were detected in the blood starting from day 4 up to day 8 post injection and presented similar dynamics. In a second set of experiments, we analyzed the phenotype of the expTregs in different organs at day 6 post injection (when expTregs were still detected in the blood). Lower CD3+ T-cell reconstitution was observed in the spleen of mice receiving WT or CD74KO expTregs compared to mice receiving PBMCs alone, indicating that both Tregs restrain xenogenic T-cell expansion in this model (Fig. S3B). No major phenotypic difference was observed between WT and CD74KO expTregs for all markers analyzed (PD1, GITR, CCR8, ICOS, CTLA-4, OX40, CD25, TIGIT, and 4.1BB) except for HLA-DR, whose expression was indeed expected to be reduced in CD74KO expTregs (Figs. 3C, D and S3C, D). Mice receiving only PBMCs developed severe GvHD (loss of weight higher than 20%) starting around day 20, while co-injection of either WT or CD74KO expTregs delayed the appearance of clinical signs (Fig. 3E, F). This protective effect of expTregs correlated with a statistically significant improved overall survival, with no evident differences between the CD74KO and WT Tregs.

Taken together, these results show that CD74-deficient Tregs control xeno-GvHD development to a similar extent as WT Tregs, suggesting that, as observed in vitro, CD74 is dispensable for Treg suppressive activity in vivo in GvHD model.

## CD74 specifically stabilizes the suppressive phenotype of Tregs within tumors

We next investigated whether in the tumor tissue—where we originally observed the surface upregulation of CD74 in Tregs—the absence of CD74 impacts Treg function. For this, NSG mice were s.c. grafted with MDA-MB231 triple negative breast cancer cells. At day ten, when tumors were palpable, mice were left untreated (control) or were injected with fresh PBMCs alone or co-injected with in vitro expanded WT or CD74KO Tregs (expTregs) (Fig. 4A). Figure 4B shows the individual (left) and the mean (right) tumor growth curves for one representative donor (donor 1). We observed that tumors grew exponentially in control mice and that injection of PBMCs led to tumor rejection. Addition of the WT expTregs slightly delayed tumor elimination. In sharp contrast, CD74KO expTregs significantly accelerated the tumor rejection. Although some inter-donor variability existed, similar results were obtained with 3 additional donors (Fig. 4C), as well as when using frozen instead of fresh expTregs (Fig. S4A). This set of experiments revealed that CD74 expression in Tregs is required for Tregs to maintain their suppressive function in the tumor. Furthermore, the enhanced tumor rejection observed with the transfer of

CD74KO expTregs suggests that in the absence of CD74, Tregs no longer suppress antitumor immune responses but may further acquire anti-tumoral functions.

The results above raised the question of whether CD74 plays a cell-intrinsic role in the control of Treg fitness and functionality specifically in the tumor. To evaluate this assumption, we analyzed the adoptively transferred Tregs in the spleen and the tumor 6 days after the injection (Fig. S4B). Remarkably, we observed a similar expTregs:PBMCs ratio for WT and CD74KO Tregs in the spleen, but a significant reduced ratio for tumor CD74KO Tregs (Fig. S4C), linking genetic deletion of CD74 in Tregs to impaired access/retention selectively in the tumor. No major phenotypic difference was found in the frequencies of WT or CD74KO splenic Tregs expressing all the analyzed molecules (PD1, CCR8, GITR, 4-1BB, ICOS, CTLA-4, OX40, and CD25) except, as consistently observed, for a reduced surface expression of HLA-DR in CD74KO expTregs (Fig. S4D, E). Geometric mean expression of these markers was also similar (Fig. S4E). In contrast, in the tumor, significantly lower frequencies of CD74KO Tregs expressing 4-1BB, ICOS, CTLA-4, OX40, CD25, and HLA-DR and higher frequencies of CD74KO Tregs expressing PD1+ were found, while Tregs positive for CCR8 and GITR were equally distributed among the two Treg populations (Figs. 4D and S4F).

To understand whether CD74 exclusively regulates the Treg suppressive phenotype in the tumor, we performed a direct comparison of the phenotype of the CD74WT and KO Tregs from spleen, liver and tumor (Fig. S4G, H). We observed that: (i) the proportion of liver Tregs expressing CD74 at the surface was elevated compared to splenic Tregs, yet lower than in tumor Tregs, and (ii) as previously described for the spleen, CD74 deletion did not induce significant changes in CD25 or 4-1BB expression in Tregs from the liver, but significantly affected Tregs from the tumor. The higher CD74 expression of liver compared to spleen Tregs, suggests that CD74 upregulation could underly an adaptive response of Tregs to residency in non-lymphoid tissue. Nevertheless, even with this increase, liver Tregs do not exhibit a clear dysfunctional phenotype (conserved CD25 and 4-1BB expression) compared to tumor tissue Tregs. This highlights a specific role for CD74 in Treg biology within the tumor environment, different from its role in non-tumor tissues.

Given that the loss of Foxp3 expression in Tregs has been previously associated to an unstable Treg phenotype[17], we analyzed the geometric mean of Foxp3 in the different conditions. The geometric mean of Foxp3 in WT and CD74KO Tregs were similar before injection (Fig. S4I), and 6 days after the transfer it was slightly lower in spleen CD74KO Tregs (Fig. 4E). Strikingly, WT Tregs show a significant increased Foxp3 level in the tumor compared to the spleen, while oppositely tumor-infiltrating CD74KO Tregs show a significant drop in Foxp3 expression (Fig. 4E). The increased Foxp3 expression in tumor WT Tregs may reflect adaptation to the harsh tumoral conditions, which seems to be impaired in the absence of CD74 expression. To further study the mechanism underlying this Treg intrinsic instability, we analyzed the consequences of in vitro stimulation of WT and

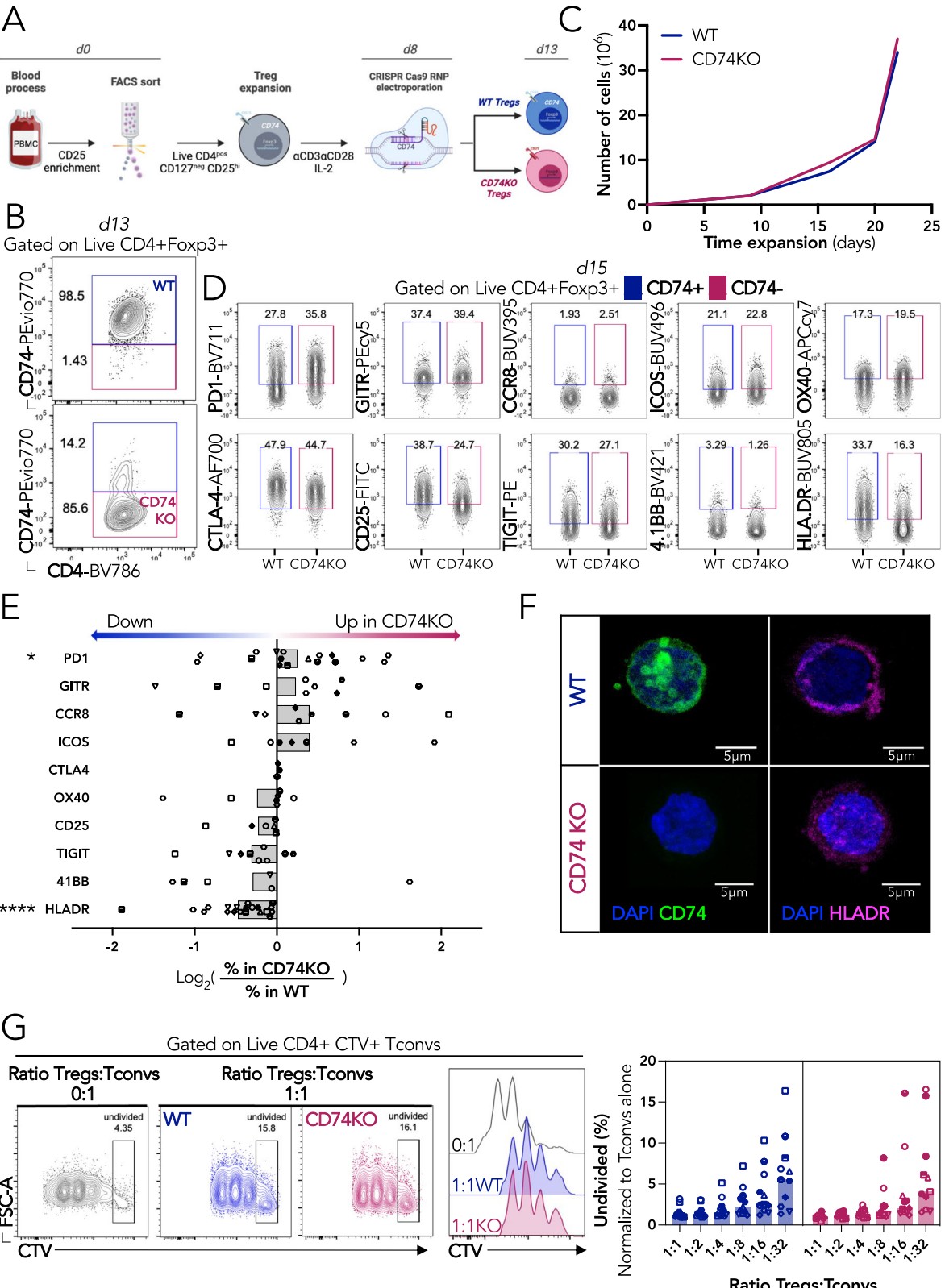

CD74KO Tregs under different stimuli capable of inducing Foxp3 downregulation and Treg destabilization[17,18]. We observed that αCD3αCD28, IFNα, IFNγ, PMA-ionomycine, and interleukine-12, induced a more pronounced loss of Foxp3 expression in CD74KO than in WT Tregs (although not statistically significant for IFNα) (Fig. S4J). These results indicate that under some inflammatory conditions, CD74 is needed to ensure Treg stability.

To better understand how CD74 could be impacting Treg maintenance in the tumor, we activated WT and CD74KO Tregs in the presence of tuSN, and/or IL-12- a cytokine known to induce Treg destabilization[19,20]. First, we quantified caspase-3 induction by FACS as a surrogate for Treg apoptosis induction (Fig. 4F). Activation of Tregs in the presence of tuSN alone or with IL-12, significantly induced apoptosis in CD74KO, but not in WT Tregs, indicating that CD74

**Fig. 2 | CD74 is not required for Treg survival, phenotype, and suppressive capacity in vitro. A** FACS-sorted Tregs are expanded for 8 days and electroporated with two CRISPR-Cas9 RNPs targeting CD74 at exon 5 and exon 6 (CD74KO) or with a control RNP (WT). Electroporated Tregs are expanded and activated for further characterization. **B** Representative dotplots of CD74 intracellular expression 5 days after electroporation (d13) for CRISPR-Cas9 deletion ($n = 35$ different donors with paired WT and CD74KO Tregs, gating strategy in Fig. S2A). **C** Proliferation curve, based on cell count, during expansion of WT and CD74KO Tregs starting after the electroporation (representative curve of one donor out of $n = 33$ different donors with paired WT and CD74KO Tregs). **D, E** WT and CD74KO Tregs are stained for Treg-activation markers 7 days after electroporation (d15) and analyzed by FACS to evaluate phenotypic changes. **D** Representative dotplots of Treg-markers expression in concatenated WT and CD74KO Tregs from the same donor (concatenation on the similar number of pure Tregs, WT, and CD74KO Tregs represent each 50%).

**E** Ratio of the percentage of positive cells for each marker in CD74KO vs WT Tregs (**D, E** $n$ represents different donors, $n = 3$ for CTLA-4, $n = 6$ for 4-1BB and CD25, $n = 7$ for OX40 and ICOS, $n = 9$ for TIGIT, GITR and CRR8, $n = 17$ for PD1). **F** Confocal images of a z-stack projection of WT and CD74KO Tregs stained for DAPI (blue), CD74 (green), and HLA-DR (magenta). Representative images of staining performed on 3 donors with paired WT and CD74KO Tregs. **G** Suppressive capacity of WT and CD74KO Tregs is assessed by co-culture with CTV-stained Tconvs at different ratios of WT or CD74KO Tregs. Dotplots and histograms of CTV staining representing the proliferation of Tconvs alone (gray), with WT Tregs (blue) or with CD74KO Tregs (red); and percentage of undivided Tconvs with different concentrations of WT Tregs (blue) or CD74KO Tregs (red) ($n = 11$, different donors with paired WT or CD74KO Tregs). Statistical analyses are performed using a paired $t$-test; with $p$ values shown as asterisk to maintain readability, p = 0,027 for PD1 (**E**). Source data are provided as a Source Data file. **A** was created with BioRender.com.

contributes to increased Treg survival in the presence of tumor-derived soluble factors. Second, we quantified IFNγ production, as a surrogate for acquisition of effector functions (Fig. 4F). Incubation with tuSN with IL-12 induced the highest IFNγ production, with the maximal increase observed in CD74KO Tregs, pointing to a crucial role for CD74 in the preservation of Treg suppressive function in the TME. Furthermore, caspase-3 induction and IFNγ production correlated with a decrease in Foxp3 geometric mean (Fig. 4G).

Additionally, we measured the methylation level of the Treg-specific demethylated region (TSDR) of Foxp3, an evolutionary conserved non-coding element present in the Foxp3 gene locus, which, when demethylated, sustains Foxp3 expression[21]. We observed that upon expansion with αCD3αCD28-coated beads and IL-2 (control condition), WT and CD74KO Tregs show similar percentages of Foxp3 TSDR demethylation (99.96% and 99.79%, respectively), and that control CD4+ T cells have a much lower Foxp3 TSDR demethylation level (0.62%) than Tregs (Fig. 4H). However, in the presence of tuSN, TSDR demethylation decreased more in CD74KO Tregs (53.33%) compared to WT Tregs (69.40%). This difference was deepened with the addition of IL-12, as Foxp3 TSDR was demethylated at 10.26% in CD74KO Tregs, against 47.57% in WT Tregs. This result suggests that CD74 plays a regulatory role in preserving TSDR demethylation and provides a molecular basis for the CD74-mediated maintenance of Foxp3 stability in Tregs.

Altogether these results reveal a previously unsuspected role for CD74 in the stabilization of Foxp3 expression in Tregs and the maintenance of their phenotype and function in the tumor.

## CD74 is specifically required for retention of Tregs within tumors

Given that CD74KO-Tregs accumulated less in the tumor than WT Tregs (Fig. S4C) and that CD74 can act as a receptor for chemokines produced by the tumor[22], we designed an experiment to more accurately quantify the contribution of CD74 deficiency on Treg infiltration of the tumor. For this, we co-injected in the same tumor-bearing NSG mice, a mix of PBMCs, and WT and CD74KO expTregs stained with different cell trace dyes and quantified the amount of WT and CD74 KO expTregs present in different organs 6 days later (Fig. 5A). Similar absolute numbers and frequencies of WT and CD74KO expTregs were found in the spleen and the liver, whereas a significantly lower proportion of CD74KO expTregs accumulated within the tumor (Figs. 5B and S5A). Indeed, only 28.67% of CD74KO expTregs were found in the tumor as compared to 50.04% in the spleen and 45.38% in the liver. These results point to a non-redundant role for CD74 in favoring Treg migration and/or accumulation specifically in the tumor.

Additionally, we used an anti-CD74 antibody (Milatuzumab) that blocks surface CD74[23] to assess the impact of CD74 surface expression on tumTregs infiltration. MDA-MB231 tumor-bearing mice co-transferred with PBMCs and WT expTregs were treated with three rounds of anti-CD74 antibody (Fig. S5B). Both intravenous and

intratumoral administration of anti-CD74 resulted in a significant reduction of expTregs in the tumor compared to the untreated mice (Fig. S5C). Conversely, a non-significant decrease in CD4 + T cells was observed, in accordance with the high and low levels of CD74 surface expression on these cells. These findings suggest that surface CD74 plays a critical role in tumTregs infiltration.

To validate this observation, we generated CD74-overexpressing Tregs (CD74OE Tregs) by transducing FACS-sorted Tregs with an intracellular-truncated CD74 mutant previously described to stabilize CD74 membrane expression by slowing down its endocytosis[24] (Fig. S5D). Then, tumor-bearing NSG mice were co-injected with PBMCs plus WT or CD74OE expTregs and analyzed 6 days later (Fig. S5E). Mirroring the observation in human samples (Fig. 1E), a higher proportion of WT expTregs expressed CD74 at the surface in the tumor compared to the spleen. Additionally, the CD74OE expTregs showed the highest tumor accumulation, validating the role of CD74 in facilitating Treg accumulation in the tumor (Fig. S5F).

To further explore the potential supplementary role of membrane CD74 as MIF co-receptor, we quantified the co-expression of CD74 with the established MIF co-receptors CD44 and CXCR4[25,26], in Tregs from spleen, liver, and tumor of MDA-MB231-bearing NSG mice (Fig. S5G). All Tregs expressed CD44 on their surface in the three tissues; and tumTregs expressed increased levels of surface CXCR4 compared to splenic and hepatic Tregs, resembling the CD74 surface expression pattern (Fig. S5H). In more details, the proportion of double-positive CD74+CD44+ and CD74+CXCR4+ is elevated in tumTregs (19.43 ± 5.35% and 13.94 ± 8.88%, respectively) and hepatic Tregs CD74 (20.00 ± 9.25% and 13.13 ± 9.19%, respectively), compared to splenic Tregs (5.62 ± 2.40% and 1.15 ± 1.15%, respectively). These results suggest that in liver and tumor, upregulation of CD74 on the surface of Tregs could be associated with its role as a co-receptor for MIF[27,28]. To investigate the potential impact of MIF ligation on CD74 on Treg stability, we incubated WT or CD74KO Tregs with soluble MIF, with or without MDA-MB231-derived supernatant, and assessed Foxp3 levels (Fig. S5I), but no discernible differences were observed. We concluded that under the tested conditions, MIF ligation on CD74 may not play a significant role in modulating Foxp3 expression and, consequently, Treg stability. Additionally, to evaluate MIF effects in vivo, we generated MDA-MB231 cell lines KO for MIF expression using a CRISPR-Cas9 RNP targeting MIF at exon 2, and the corresponding control MDA-MB231 cells using a control RNP. We verified the efficacy of MIF targeting by measuring MIF concentration by ELISA in the supernatant of the tumor cells 4 days after MIF deletion and observed that although MIF concentration was significantly reduced, it was not completely abrogated at this time point (Fig. S5J). Control (MIF+) and MIFKO tumor cells were grafted in NSG mice, and 10 days later we co-transferred in the same mouse a mix of PBMCs, WT, and CD74KO Tregs (Fig. S5K). We observed that both tumors developed similarly in NSG mice, indicating that

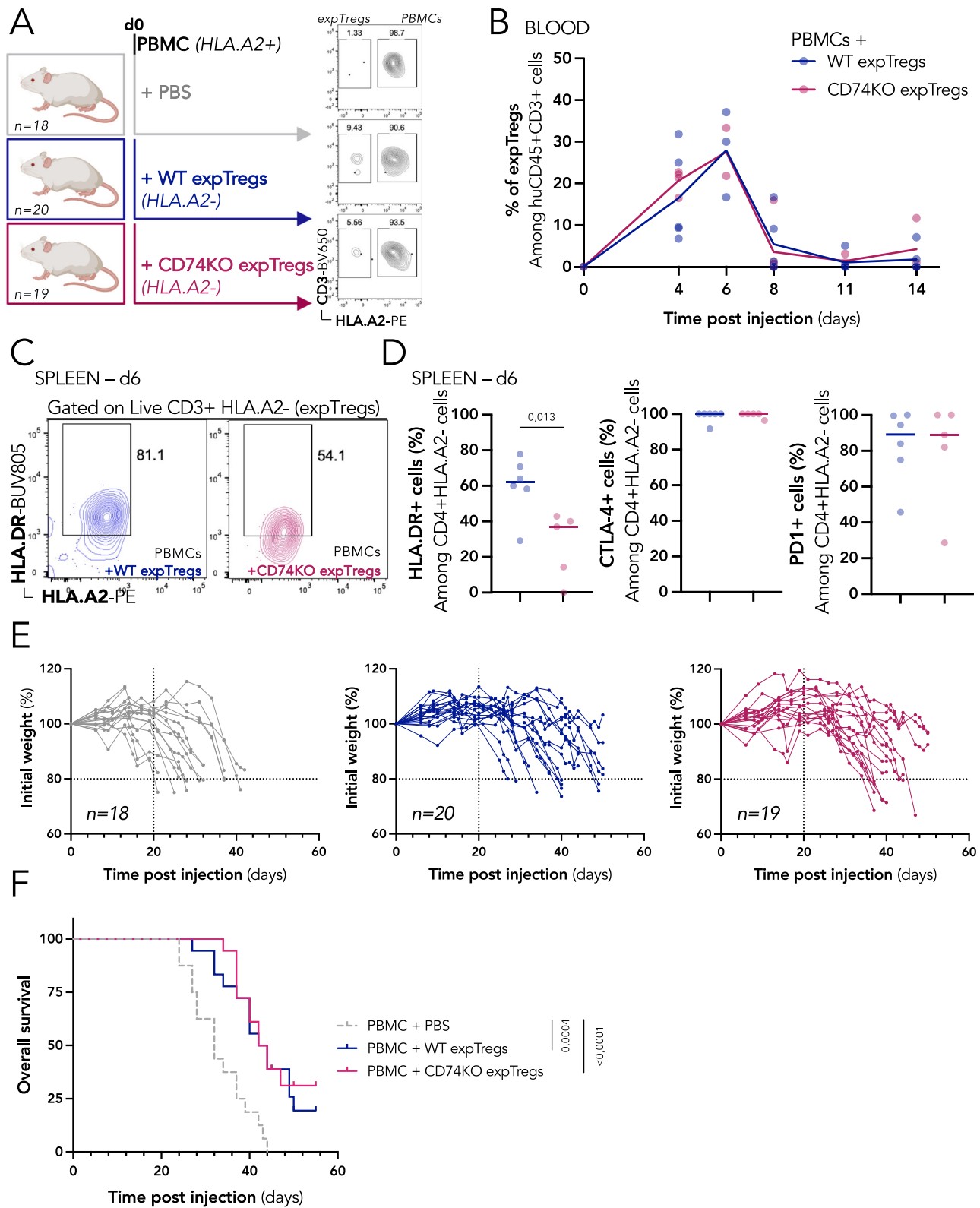

**Fig. 3 | CD74-deficient Tregs conserve their suppressive function in vivo in a xeno-GvHD model. A** To induce xeno-GvHD, PBMCs (HLA.A2+) are injected intravenously in immunodeficient NSG mice alone (gray, *n* = 18) or together with WT (blue, *n* = 20), or CD74KO (red, *n* = 19) expanded Tregs (expTregs, HLA.A2-). Representative dotplots of PBMC-expTreg proportion among huCD45 + CD3+ cells. **B** Detection of WT (blue, *n* = 5) and CD74KO (red, *n* = 5) expTregs in the blood of mice injected with PBMCs and expTregs at early timepoints, percentage of HLA.A2- cells among huCD45 + CD3+ cells by FACS. **C, D** Splenocytes are stained for activation markers of Tregs 6 days after cell injection. Representative dotplots of

HLA.DR surface expression in WT and CD74KO expTregs (gating strategy in Fig. S3A). **C** and quantification of cells expressing HLA.DR, CTLA-4, and PD1 in WT or CD74KO expTregs from *n* = 6 or *n* = 5 mice, respectively (**D**). **E** Individual weight-loss curves for the three groups of mice, calculation based on the initial weight of mice at the day of T-cell injection. **F** Survival curves for the same 3 groups of mice. Statistical analyses are performed using an unpaired *t*-test (**D**); or using a Mantel-Cox test (**F**); *p* values are shown on the graphs. Source data are provided as a Source Data file. **A** was created with BioRender.com.

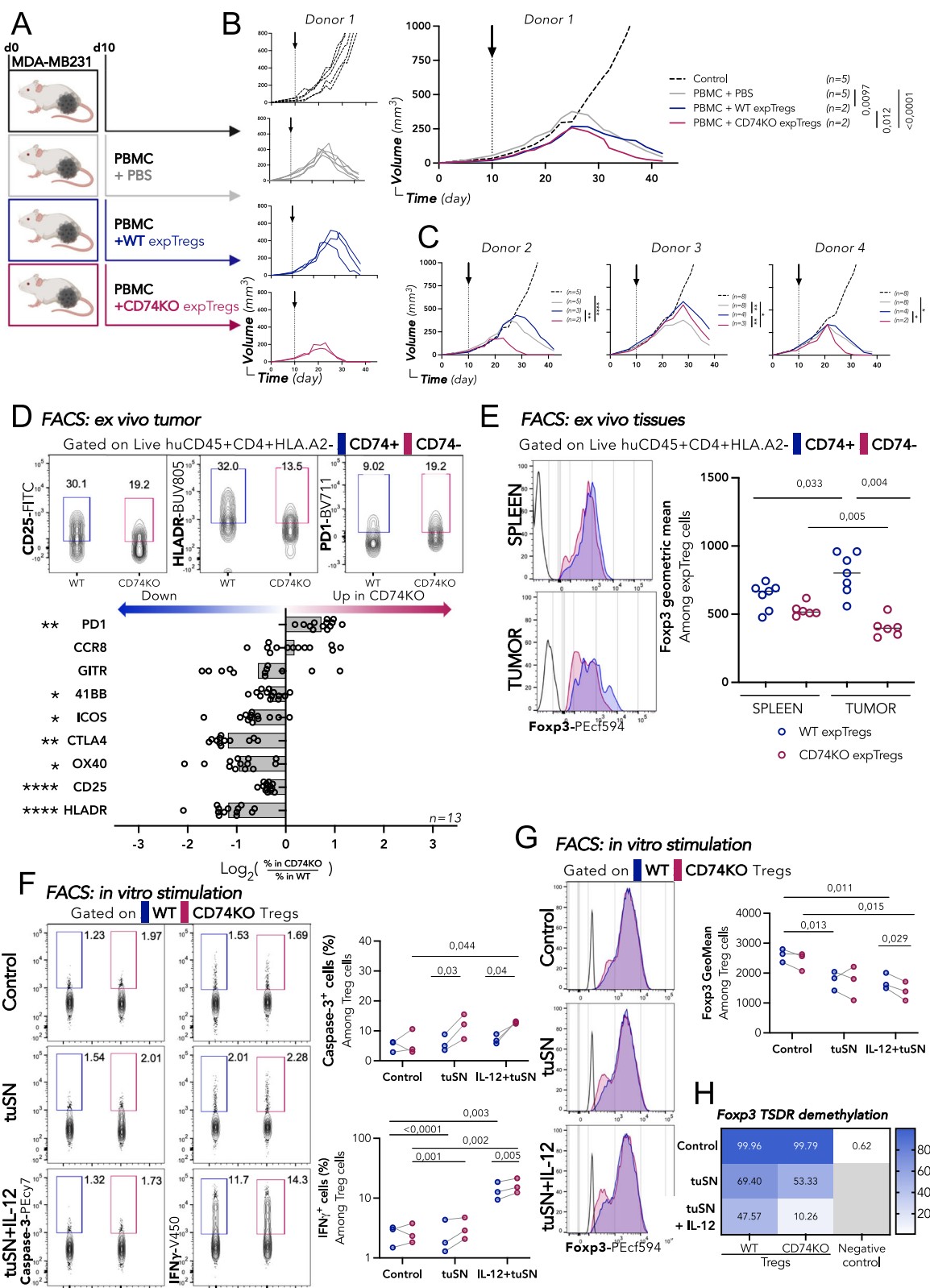

**D** *FACS: ex vivo tumor*

Gated on Live huCD45+CD4+HLA.A2-

**E** *FACS: ex vivo tissues*

Gated on Live huCD45+CD4+HLA.A2-

**F** *FACS: in vitro stimulation*

Gated on WT / CD74KO Tregs

**G** *FACS: in vitro stimulation*

Gated on WT / CD74KO Tregs

**H** *Foxp3 TSDR demethylation*

MIFKO did not affect per se the tumor growth capacity. Seven days after transfer, we quantified tumor Tregs. As in the previous experiments, higher amounts of WT Tregs were observed compared to CD74KO Tregs in both MIF+ and MIFKO tumors with no major differences between the two tumors (Fig. S5L–N). While the remaining MIF production by MIFKO tumor cells might mask a potential impact of MIF on Treg recruitment, these findings do not suggest a major role for CD74 in the response of Tregs to MIF. Overall, although MIF co-receptors are expressed in tumTregs, these series of experiments do not support a major role for MIF ligation on CD74 in contributing to Treg stability in the tumor.

Overall, our results indicate that CD74 plays a non-redundant and specific role in Treg recruitment and/or retention within tumors but seems not to be mediated by CD74 function as MIF co-receptor.

**Fig. 4 | CD74 promotes activation and accumulation of Tregs specifically in the tumor. A** MDA-MB231 tumor cells are engrafted subcutaneously in the flank of immunodeficient NSG mice. 10 days later (when tumors are palpable), HLA.A2+ PBMCs are injected intravenously alone (gray) or together with WT (blue), or CD74KO (red) in vitro expanded HLA.A2- Tregs (expTregs), a group of mice is not injected as a control (black). **B** Individual (left) and median (right) tumor-growth curves for each group of mice injected with one representative donor of expTregs. **C** Median of tumor-growth curves with 3 additional donors. **D, E** 6 days after T-cell injection, splenocytes, and tumor-derived cells are stained for activation markers of Tregs. **D** Representative dotplots of CD25, HLA-DR and PD1 surface expression in concatenated samples of WT ($n = 7$) and CD74KO ($n = 6$) tumor expTregs (top), (gating strategy in Fig. S4B), and quantification of the markers in expTregs, represented as the ratio among CD74KO versus WT expTregs (bottom). **E** Representative histograms (left) and quantification (right) of Foxp3 level in WT (blue) ($n = 7$) or CD74KO (red) ($n = 6$) expTregs from spleens and tumors recovered 6 days after T-cell transfer. **F–H** CD74KO and WT Tregs ($n = 3$ donors) were cultured with αCD3αCD28-coated beads and IL-2 (1000 IU/mL) for 72 h (control), and further cultured with MDA-MB231 derived supernatant (tuSN) alone, or together with IL-12 (20 ng/mL) for 24 h before analysis. **F** Representative dotplots (left) and quantification (right) of Caspase−3 and IFNγ expression in WT or CD74KO expanded Tregs. **G** Representative histograms (left) and quantification (right) of Foxp3 level in WT or CD74KO expanded Tregs. **H** Percentage of Foxp3 TSDR DNA methylation in WT or CD74KO expanded Tregs, and CD4+ cells (negative control). Statistical analyses are performed using an unpaired $t$-test (**B–E**) and paired multiple $t$-test (**F, G**). $p$ values are shown on the graphs except when asterisk maintain readability with $p$ Value <*:0,1; **:0,01; ***:0,001; ****:0,0001 (**C, D**). For **D**, exact $p$ values are: PD1 (0,021), CCR8 (0,69), GITR (0,11), 4-1BB (0,04), ICOS (0,01), CTLA4 (0,0013) and OX40 (0,009). Source data and exact $p$ values (**C**) are provided as a Source Data file. **A** was created with BioRender.com.

## CD74 conditions the migratory capacity of Tregs

The striking effect of CD74 to sustain Treg retention and function selectively in the tumor prompted us to evaluate whether these observations could be triggered by a role of CD74 in Treg motility. We investigated the velocity and migratory behavior of Tregs by two types of microfabricated devices: microchannels and micropillars. Microchannels were designed to mimic the confined environment of peripheral tissues but in a one-dimension model, which facilitates the extraction of migration parameters from a large number of cells. DAPI-stained CD74 and WT Tregs were loaded in 4 µm × 5 µm microchannels and imaged for 16 h (Fig. 5C). We observed that CD74KO Tregs were impaired in their ability to enter microchannels and presented enhanced speed fluctuations (Fig. 5D). The speed fluctuation of CD74KO Tregs was twice higher than WT ones (mean of 2.45 ± 4.45 and 1.08 ± 0.86, respectively). Moreover, while WT Tregs migrated in a continuous manner within microchannels, CD74KO Tregs showed less stability in their trajectories, often changing direction (Fig. 5E). In agreement with these results, we noticed that the migration speed of CD74KO Tregs was faster when starting the experiment but decreased at later time-points (Fig. S5O). Accordingly, measurement of cell density in the channels revealed that CD74KO Tregs accumulated within the first 200 µm, but this behavior was not observed in WT Tregs that exhibited a stable spatial distribution all along the assay (Fig. 5F). These results indicate that CD74 contributes to the migratory capacity of Tregs.

Analysis of Treg migration in a forest of micropillars, which allows studying confined migration in two dimensions, showed that CD74KO and WT Tregs displayed distinct cell migration patterns. DAPI-stained CD74KO and WT Tregs were loaded in 5 µm-high pillar chambers and imaged every 2 min for 16 h (Fig. 5G). We analyzed the center-of-mass position of every cell and reconstituted their path in X-Y coordinates (Figs. 5H and S5P). To provide a robust quantification of this phenotype, we measured the "turning" angle as defined by the cell position between 3 consecutive timepoints (α, Fig. S5Q). Cell trajectories were considered as persistent when changes of direction displayed an angle $\alpha < 30°$, or as confined when $\alpha > 30°$. This analysis showed that CD74KO and WT Tregs exhibited similar track lengths (Δmean = 44.1 ± 33.3 µm), indicating that they migrate through similar distances (Fig. 5I). However, CD74KO Tregs spent 14.3 ± 2.7% of their motile time with a confined trajectory, against 11.9 ± 3.0% for the WT Tregs, meaning that CD74KO Tregs change direction more frequently as compared to WT Tregs (Fig. 5I). Of note, this phenotype was totally unexpected as the opposite effect of CD74 has been described on DC migration[5].

Taken together our results show that CD74 deficiency disrupts Treg migration by increasing their speed fluctuations and thereby reducing the directionality of their migration trajectories, consistent with our in vivo data highlighting that CD74 KO Tregs fail to accumulate within the tumor environment.

## CD74 deficiency in Tregs leads to transcriptional changes related to intracellular organelles and cytoskeleton

To understand the molecular mechanisms underlying the effect of CD74 on Treg motility and tumor retention, we profiled WT and CD74KO Tregs using bulk RNA sequencing. Cells were generated from three HD PBMCs as in Fig.2A. KO efficacy was confirmed at the protein level by FACS analysis, as well as at the transcriptomic level, by the decrease in the number of reads that aligned with the two CD74-targeted regions (Fig. S6A, B). We identified 25 differentially expressed genes (DEGs) overexpressed in WT Tregs and 148 in CD74KO Tregs (Fig. S6C). A high proportion of the genes overexpressed in WT Tregs was associated with microRNA and RNA processes, such as mRNA splicing and mRNA binding activity pathways. These genes included small nucleolar RNA genes like SNORA −5A, −38, −75; as well as the RNA components as RNU4 and RNU5 (Figs. 6A and S6D). At odds, genes overexpressed in CD74KO Tregs, were associated to cell morphogenesis and extracellular matrix, such as keratin, collagen, protocadherin; as well as to intracellular organelle processes, with the example of the golgin family like genes *KRTAP10, COL6A5, PCDHA4, GOLGA8CP*; and to ion binding and transport, like the *KCNJ12* and *SLC7A10* genes (Figs. 6A and S6E). The top10 upregulated cellular-component terms for WT Tregs (from GSEA analysis) were associated to proteasome regulation and RNA complex interaction, suggesting a role for CD74 in the upregulation of RNA activity (Fig. 6B), and in accordance with the described ability of CD74 intracellular domain to act as a transcription regulator[29]. The DEGs overexpressed by CD74KO Tregs were associated to terms all linked to cell structure and membrane interactions, including cell morphogenesis (containing more than half of the DEGs upregulated in CD74KO Tregs), cell-cell junction, ion binding, and ion transport (Fig. 6C). This observation was confirmed using EnrichR libraries, which identified among the top 25 processes upregulated in CD74KO Tregs, pathways related to cell structure (Golgi membrane, cytoskeletal fiber, and intermediate filament), ionic homeostasis (sodium and phosphate) and membrane biology (brush borders and synapse assembly pathways) (Fig. S6F). We observed that for the GSEA pathways "membrane structure", "cellular organelles", "cellular matrix", and "adhesion ability", all the DEGs were overexpressed in CD74KO Tregs compared to WT Tregs, supporting the implication of CD74 in cell morphogenesis, structure, and function (Fig. S6G).

These results are consistent with the ability of CD74 to regulate membrane trafficking and cytoskeleton organization in APCs[5,30].

## CD74 triggers the reorganization of the Treg's shape and intracellular organelles

Guided by the transcriptome results, as well as the impact of CD74 in Treg motility, we analyzed the shape and distribution of organelles in CD74KO Tregs. After 7 days of culture, we stained WT and CD74KO Tregs from 3 donors with DAPI, anti-CD74, phalloidin (an actin cytoskeleton marker), anti-Rab6 (a Golgi marker), and anti-EEA1 (an early-

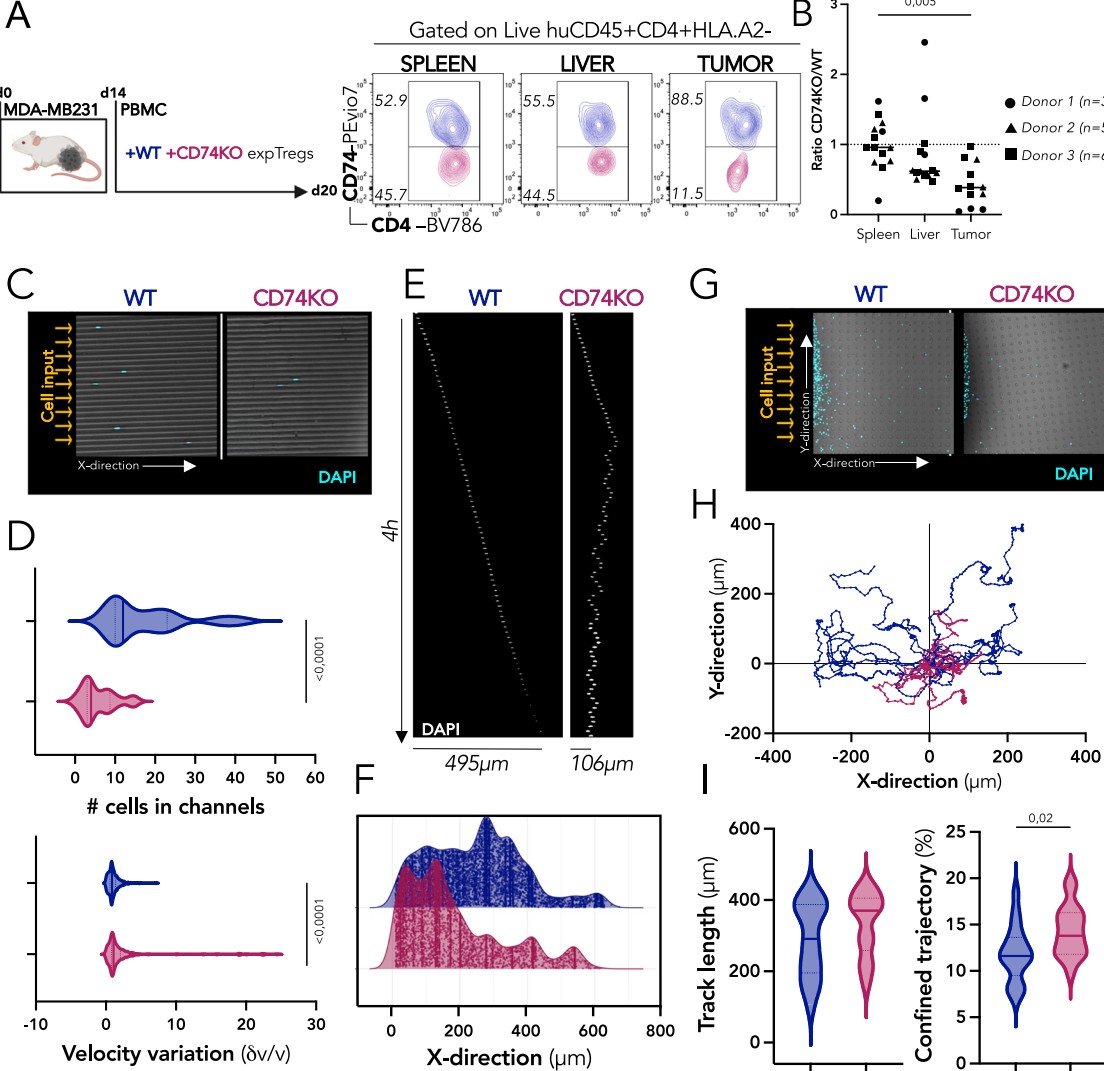

**Fig. 5 | CD74 stabilizes Treg motility by modulating cell speed and maintaining a diffused migration. A** MDA-MB231 tumor cells are engrafted subcutaneously in the flank of immunodeficient NSG mice, and 14 days later, PBMCs, CFSE⁺ WT expTregs, and CTV⁺ CD74KO expTregs are injected intravenously. Six days later, spleens, livers, and tumors are analyzed by FACS. The representative dotplots for each tissue display the proportion of WT cells (blue) and CD74KO cells (red) among total expTregs (gating strategy in Fig. S4B). **B** Ratio of numbers of CD74KO versus WT cells among expTregs in spleen, liver, and tumor. **C–I** FACS-sorted Tregs from 3 donors are expanded for 7 days and electroporated with two CRISPR-Cas9 RNPs targeting CD74 (CD74KO) or one control RNP (WT). Electroporated cells are expanded and activated for 14 days and WT and CD74KO Tregs are loaded into microchannel or micropillar gels. **C** Examples of images on a 10× objective of microchannels loaded with DAPI-stained WT and CD74KO Tregs. **D** Quantification of the number of cells entering microchannels (top) and variation in speed (bottom) of WT and CD74KO Tregs. **E** Sequential images of cell displacement in a channel (one row for each frame, $t = 3$ min). **F** Cell density in the microchannels, calculated by the further distance reached by every cell. **G** Examples of images on a 10x objective of micropillar gels loaded with DAPI-stained WT and CD74KO Tregs. **H** Trajectory of the top16 longest trajectory of WT (blue) and CD74KO (red) Tregs. **I** Quantification of the trajectory length of Tregs (left) and percentage of time spent in confined displacement of each cell (right). **B** Statistical analyses are performed using a paired *t*-test and *p* value is shown on the graph. **C–I** Pooled data of the 3 donors, statistical analyses are performed using an unpaired *t*-test and *p* values are shown on the graphs. Source data are provided as a Source Data file. **A** was created with BioRender.com.

endosome marker) (Figs. 6D and S7B). The absence of CD74 detection in CD74KO Tregs validated the efficacy of the genetic deletion (Fig. 6D). Phalloidin staining highlighted that the cell inner surface of CD74KO Tregs was considerably decreased: estimated Treg inner surface was of $37.87 \pm 25.28\,\mu m^2$ for WT cells and of $24.21 \pm 19.92\,\mu m^2$ for CD74KO cells (Fig. 6E). CD74KO Treg smaller size was confirmed by live imaging using the holotomography technology that allowed to study cells in suspension (Fig. S7D, E). This observation suggests that CD74 controls Treg size and cytoskeleton contractility, as previously shown for DCs[5].

Phalloidin staining further revealed a distinct abundancy of membrane protrusions between WT and CD74KO Tregs. Although

the length of the actin spikes was not significantly different, their number was increased at CD74KO Treg surface (Figs. 6F and S7A). In addition, using electronic microscopy, we observed a similar hairy structure of the CD74KO Tregs and validated that they possess significant higher number of membrane protuberances (Fig. 6H, I).

Noticeably, three-dimensional image analysis showed that the volume occupied by intracellular compartments was altered in Tregs lacking CD74 (Fig. 6D). While the volume of early endosomes (EE, EEA1 staining) was increased in the absence of CD74, the Golgi apparatus (Rab6 staining) showed the opposite tendency (Fig. S7B, C). These results are consistent with data obtained in other cell types

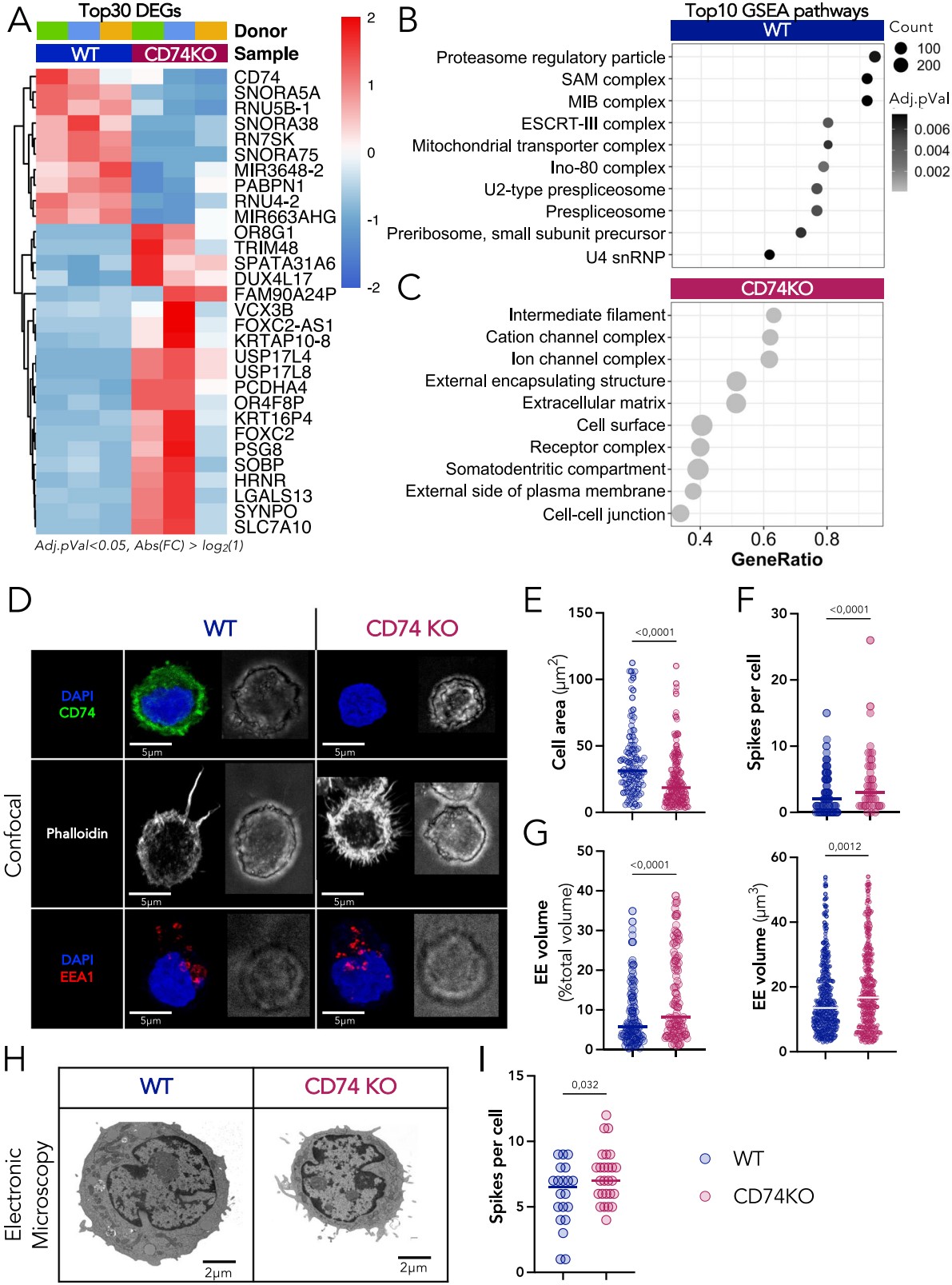

showing that CD74 controls membrane trafficking along the endocytic pathway[30]. As CD74KO Tregs were found to be smaller than WT Tregs, we calculated the percentage of the intracellular space occupied by these compartments. No difference was observed for the Golgi apparatus, while higher occupancy was found for EE in CD74KO Tregs (Figs. 6G and S7C). The holotomography technology confirmed the reorganization of Treg organelles caused by the CD74 deficiency as CD74KO Tregs displayed a higher total dry mass (Fig. S7F). We conclude that CD74 controls the morphology of Tregs at the level of cell shape and intracellular organization.

Altogether, our findings highlight that CD74KO Tregs exhibit altered morphogenesis, providing putative mechanisms to explain their failure to migrate, accumulate, and remain stably active within the TME.

**Fig. 6 | CD74 contribution to the transcriptome and cellular morphogenesis of Tregs. A**–**C** FACS-sorted Tregs from 3 donors are expanded for 7 days and electroporated with two CRISPR-Cas9 RNPs targeting CD74 (CD74KO) or one control RNP (WT). Electroporated cells are expanded for 14 days and the RNA from WT and CD74KO Tregs are harvested to perform a bulk-RNA sequencing **A** Supervised clustering heatmap showing the expression of the 30 most differentially expressed genes for each donor of Tregs, CD74KO versus WT condition. **B**, **C** GSEA was performed to assess the specific enrichment of cellular-components in CD74KO compared to WT Treg samples. Bubble plots display the top-10 enriched terms of WT Tregs (**B**) and CD74KO Tregs (**C**). The gene list of the differential expressed genes was loaded into the function: gseGO, from the clusterProfiler package. **D** Confocal images showing the localization of DAPI (blue), CD74 (green), Phalloidin (gray), EEA1 (red) in a z-stack projection of distinct WT and CD74KO Tregs and the corresponding bright-field image. **E** Measure of the inner-membrane

surface based on a z-stack projection with the phalloidin staining. **F** Count of the number of spikes per cell based on the phalloidin staining. **G** Percentage of volume occupied by early endosomes in the cell and total volume of early endosomes, analyzed with the EEA1 staining for early-endosome volume and the phalloidin staining for cell volume. **H**, **I** WT and CD74KO Tregs from 3 donors are obtained as described previously, cells are fixed to coverslips and analyzed under an electronic microscope. **H** Representative images for WT and CD74KO Tregs. **I** Count of the number of protuberances observed on WT and CD74KO Tregs. **B** The *p* value cutoff was at 0.05 and no adjustment method was applied. **E**–**I** Pooled data of 3 donors with paired WT and CD74KO Tregs. For every donor, 5 images were captured with a number of cells varying from 10 to 50 cells per image. The analysis was done by doing a mask of every cell based on the Phalloidin staining. Statistical analyses are performed using an unpaired t-test; *p* values are shown on the graphs and horizontal lines represent median (**E**–**I**). Source data are provided as a Source Data file.

## CD74 favors infiltration into tumors and maintain Foxp3 level in mouse Tregs

We set up to validate our initial observation and conclusions obtained with human cell in syngeneic mouse models. First, using immunocompetent C57BL/6 mice grafted with B16-F10 melanoma cells, we observed that as in humans, the surface expression of CD74 was higher in tumor-Tregs compared to peripheral Tregs, and to tumor and peripheral CD4+ Tconvs and CD8+ T cells (Fig. 7A, B). Additionally, the vast majority of Tregs expressed CD74 intracellularly, although not at 100%. Second, we designed a model of adoptive cell transfer to evaluate the capacity of CD74KO and WT Treg to infiltrate the tumor in the same host mice (Fig. 7C). We co-injected into MCA tumor-bearing Rag-KO mice CD4+ CD25high Tregs sorted from CD45.1 WT or CD45.2 CD74 full KO mice. After 6 days, we evaluated the infiltration of Tregs in the spleens and tumors by FACS. We observed that the KO/WT Treg ratio was slightly lower in the tumor compared to the spleen, and this was accompanied by a significant drop in the levels of Foxp3 (Fig. 7D). These results point to a potential role for CD74 in the accumulation of Tregs in the tumor and are in agreement with the previous observations in mice and men. However, as CD74KO cells were obtained from CD74 full KO animals, where the repertoire of T cells could be biased by the absence of CD74 in thymic APCs and epithelial cells[31,32], these results should be the interpreted with precaution.

To circumvent the cell-extrinsic effects of the CD74 full KO mice, we generated Foxp3-specific CD74 conditional knockout mice (CD74cKO mice) (see Methods section for details, and Fig. S8A–F). To analyze the impact of CD74 deletion on tumor Tregs, CD74 control and CD74cKO mice were treated with tamoxifen (on day 0, 7, and 14), grafted with MCA or MC38 tumor cells on day 7, and tissues were analyzed on day 18 by flow cytometry (Fig. 7E). In both tumor types, total CD74 was expressed in an important proportion of tumor-Tregs, and at lower levels in Tregs from other tissues, and in CD4+ Tconvs and CD8+ T cells in the tumor (Fig. S8G, H). Tamoxifen administration induced a significant deletion of CD74 mainly in Tregs in all tissues, in a similar pattern for both tumor types (Fig. S8G, H). Although there was a tendency to a lower ratio of CD74KO vs control Tregs in the tumor compared to the spleen in the MCA model (Fig. 7F, left panel), no differences were observed for the MC38 tumors (Fig. 7G, left panel), with a very high dispersion in the tumor. However, a consistent significant drop in the level of Foxp3 expression was detected in the Tregs in both tissues, and for the two tumor models analyzed (Fig. 7F, G, right panels). These data reinforce the working hypothesis that CD74 is upregulated by Tregs in the tumor, and that it contributes to the maintenance of Foxp3 stability.

Altogether, results obtained in syngeneic mouse models validate our initial observation that, as in humans, CD74 plays an important role for the adaptation of Tregs to the TME and for the maintenance of Foxp3 stability.

## Discussion

Tregs represent a main barrier to tumor immune rejection, as they induce tumor immune tolerance and promote tumor angiogenesis[33]. Consequently, selectively depleting or inhibiting Tregs in the tumor and sparing Tregs in healthy tissues -needed for the maintenance of immune homeostasis- should enable tumor elimination, especially if combined with other immunotherapies. Tumor-Treg targets may derive from Tregs' unique molecular reprogramming endowing their adaptation to the highly hypoxic and poor nutrient content conditions of the TME, mediated in part by Foxp3[34,35]. In the tumor, Tregs are found in an activated state, expressing high levels of CD25, CTLA-4 and HLA-DR, and other markers, such as 4-1BB, OX40, ICOS, LAG3 or CD49b; and the expression of these markers has been shown to correlate with a heightened suppressive capacity[36–38]. In this study, we revealed that in the tumor, CD74 is upregulated at the membrane of Tregs, but not of Tconvs or CD8+ T cells (Figs. 1D and S1C, D), and that distinct CD74 expression in Tregs and Tconvs is partly driven by epigenetic factors (Fig. 1F). We initially hypothesized that CD74 upregulation in Tregs was directly linked to Treg activation, as part of their role as chaperon of the activation marker HLA-DR. However, we were unable to induce membrane upregulation of CD74 on primary Tregs in vitro, neither upon classical TCR-mediated activation, nor in the presence of inflammatory cytokines, such as IFNα, IFNγ or tumor-derived supernatant (Fig. S1F), indicating that CD74 stabilization on the membrane was unique to the TME. Furthermore, CD74 cell-intrinsic role was dissociated from Treg activation and suppressive function, at least in vitro or during GvHD in vivo, where genetic KO of CD74 in Tregs did not induce visible changes compared to their WT counterparts. Strikingly, it was uniquely in the tumor where CD74KO Tregs showed impaired accumulation, unstable phenotype and loss of Foxp3 expression. Indeed, in NSG mice bearing human tumors and transferred with PBMCs and WT or CD74KO Tregs, the few CD74KO Tregs retained in the tumor showed significant lower expression of activation markers and a drop in Foxp3 level, while WT and CD74KO Tregs from the spleen were found in equal numbers and showed a stable phenotype. Accordingly, mice transferred with CD74-deficient Tregs showed accelerated tumor rejection, likely explained by CD74KO Treg conversion into effector-like T cells. Similar findings were obtained in syngeneic mouse models, reinforcing our initial observation CD74 plays an important role for the adaptation of Tregs to the TME and for the maintenance of Foxp3 stability.

Comparison of the transcriptome of WT and CD74KO Tregs gave interesting insights into the molecular changes induced by the genetic deletion of CD74. One intriguing observation was the association of CD74 with "morphogenesis" and "organelle trafficking" pathways. These inferred cellular modifications were confirmed by microscopy observation of the cells, which unveiled that CD74KO Tregs had a smaller size and a perturbed actin organization compared to WT Tregs and showed a characteristic hairy shape due to the development of large spikes at the membrane. Mechanistically, CD74 has been described to interact with SNAREs and actin-related proteins, and to regulate

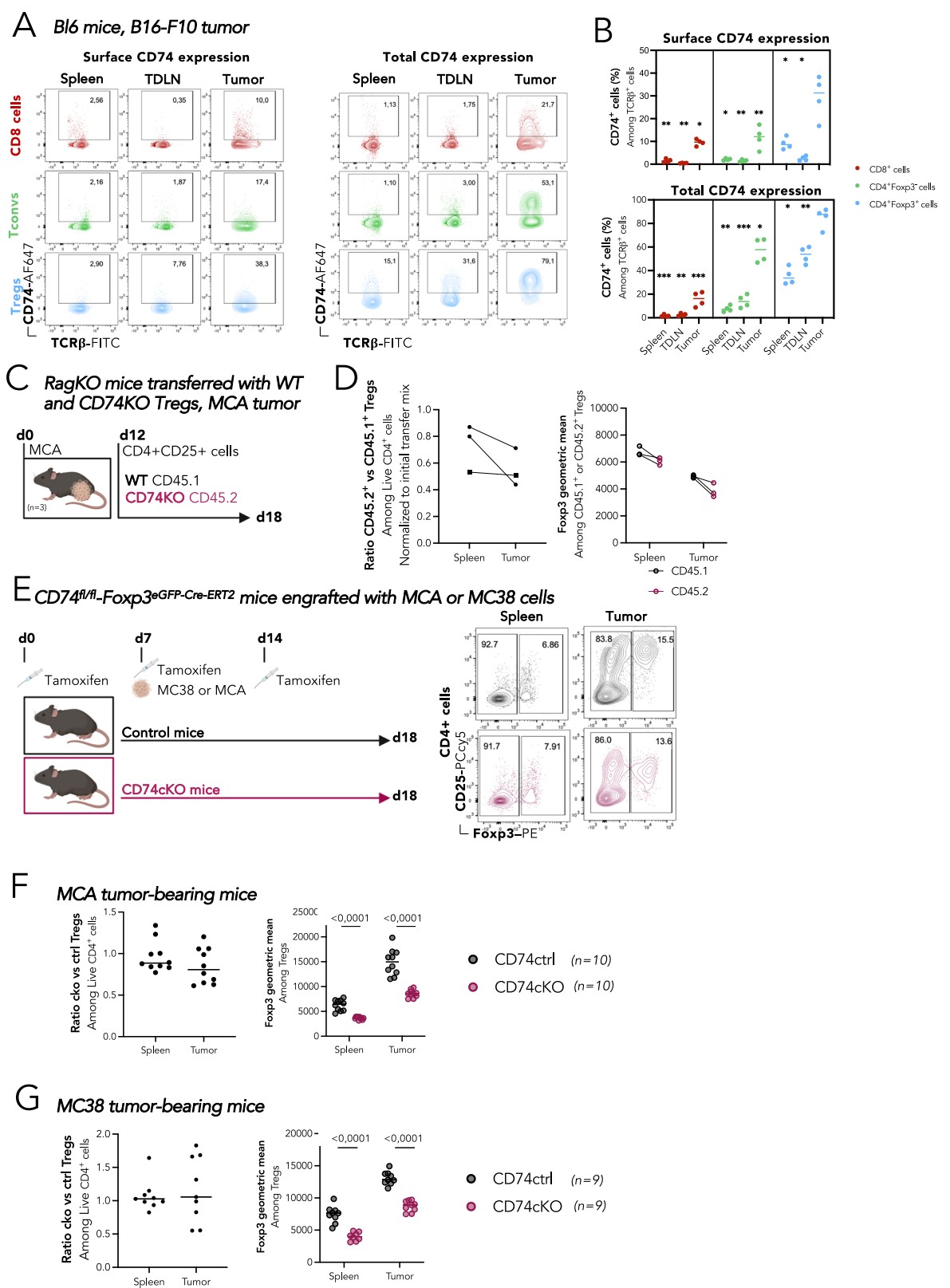

the fusion and trafficking of intracellular compartments in APCs. As previously showed in a human melanoma cell line endogenously expressing MHC-II[30], we observed that CD74 deficiency led to an increase in the total load of early-endosomes in Treg cytosol. Additionally, the implication of CD74 in the conservation of the Treg cytoskeleton and matrix may underlie the impaired motility and accumulation of CD74KO Tregs observed in our live imaging studies. In this

sense, comparative studies of mouse *Cd74-/-* or WT DC have previously identified CD74 as a regulator of DC motility[5]. Thus, impaired retention and activation of CD74KO Tregs in the harsh conditions imposed by the TME seems to arouse as a consequence of the essential role of CD74 in maintaining the stability of Treg membrane border, cytoskeleton structure, and early-endosome recycling, and thus conditioning Treg locomotion and cell-cell interaction with the tumor matrix.

**Fig. 7 | CD74 favors infiltration into tumors and maintain Foxp3 level in mouse Tregs. A, B** C57BL/6 mice ($n = 4$) were engrafted with $0.5 \times 10^6$ B16-melanoma cells, and 3 weeks later, spleens, tumor-draining lymph nodes and tumors were analyzed by FACS. Representative FACS dotplots among live TCRβ+ cells (**A**) and quantification (**B**) of % of cells expressing CD74 at the surface (**A** left, **B** top) or total (**A** right, **B** bottom) in TCRβ + CD8+ T cells (red), TCRβ + CD4+ Foxp3- Tconvs (green) and Foxp3+ Tregs (blue). **C, D** RagKO mice ($n = 3$) were engrafted with $0.5 \times 10^6$ MCA cells, and 12 days later, a mix of CD4 + CD25+ enriched splenocytes from CD45.1 WT and CD45.2 CD74KO mice was injected i.v. Six days later, spleens and tumors were analyzed by FACS (**C**) On the left, graph shows the CD45.2+/CD45.1+ Treg cell ratio in spleen or tumor, normalized to the CD45.2+/CD45.1+ Treg cell ratio of the initial transfer mix. On the right, graph shows the geometric mean of Foxp3 among CD45.2+ and CD45.1+ Tregs in spleens and tumors (**D**). **E** CD74ctrl or CD74cKO mice were treated with tamoxifen (on day 0, 7, and 14), grafted with murine tumor (MCA or MC38) on day 7 and tissues were analyzed on day 18 by flow cytometry. Ratio of the numbers of Tregs in CD74cKO mice versus the mean of the numbers of Tregs in CD74ctrl mice in spleen and tumor (left); and quantification of Foxp3 level (right) among Tregs in MCA-bearing mice (**F**) and in MC38-bearing mice (**G**). Statistical analyses are performed using a paired $t$-test with tumor-Treg value; $p$ values are shown as asterisk to maintain readability with $p$ value <*:0,1; **:0,01; ***:0,001; ****:0,0001 (**B**); and a paired $t$-test comparing control to KO condition with $p$ values shown on the graphs (**D–G**); horizontal lines represent median (**B–G**). Source data and exact $p$ values (**B**) are provided as a Source Data file. **C, E** were created with BioRender.com.

Additionally, transcriptomic data revealed that CD74 deficiency was associated to the downregulation of genes related to RNA features with more than 75% of downregulated genes belonging to the small-RNA molecule families like SNORA, lncRNA and snRNA. Although CD74 itself does not bear a DNA-binding motif, a CD74-truncated fragment was described to translocate the nucleus and interact with transcription factors like NF-kB, Bcl2 or Bxcl in B cells[4,29,39]. Thus, the intracellular domain of CD74 could also play a similar role in Tregs, modulating their transcriptional activity.

Upon repeated stimulation or under inflammatory conditions, Tregs can be destabilized, with the demethylation of the Foxp3 CNS2 locus[17], or fragilized with the loss of suppressive capacity. Both destabilized and fragile Tregs display a drop in Foxp3 expression, associated to the loss of inhibitory function, and the acquisition of effector function. Likewise, we observed that highly activated Tregs in the tumor showed reduced Foxp3 geometric mean levels, compared to the ones in the spleen; and that this loss was accentuated in CD74KO Tregs (Fig. 4E). Along these lines, when WT and CD74KO Tregs were activated in vitro in the presence of tuSN and/or IL-12- a cytokine known to induce Treg destabilization, CD74KO Tregs showed higher levels of apoptosis and of IFNγ production with a paralleled decrease in Foxp3 protein expression. These findings suggest that CD74 likely sustains Treg stability in the tumor in a cell-intrinsic fashion, function that could be ensured through the regulation of the activity of transcription-factor complexes.

Overall, these observations suggest that CD74 deficiency would decrease Treg accumulation and enhance the Treg-to-Teff conversion in the tumor by regulating Treg membrane recycling, cytoskeleton and cell junction; and/or by modulating Treg transcriptional activity. The obtained results raise the question whether CD74 expression could represent a potential therapeutic target to modulate tumor Tregs. CD74 has already been targeted by Milatuzumab, a humanized IgG1k anti-CD74 blocking monoclonal antibody, designed to target B cells in lymphoma. However, the off-target effect on lymphoma cells of this drug triggered toxicity[40]. In our model of tumor-grafted NSG mice transferred with human expTregs and PBMCs, intra-tumoral and intra-venous administration of Milatuzumab resulted in a reduction of expTregs but not in CD4+ Tconvs in the tumor (Fig. S5B, C), reinforcing the interest in CD74 as a tumor-Treg target. As for its role in tumor accumulation, CD74 manipulation to regulate T-cell retention has not yet been explored. Diminishing CD74 surface expression could be envisioned to inhibit Treg accumulation in the tumor. In an opposite strategy, increasing CD74 expression could be used to enhance Treg infiltration in inflamed tissues. Along these lines, different CD74 isoforms have been described to accumulate at a different level at the cell membrane, shorter length of the CD74 intracellular fragment correlating with a higher stabilization at the membrane. As showed by Bakke and Dobberstein[24], we observed that a mutated CD74 molecule, with a 23-amino-acid shorter cytosolic tail, resulted in an increased stabilization of its expression at the plasma membrane (Fig. S5D), and that the membrane-CD74 stabilized Tregs accumulate preferentially in the tumor (Fig. S5E, F). Thus, the engineering of Tregs transduced with a similar mutated CD74 could boost their retention in inflamed tissues in the case of autoimmune diseases, provided that inflamed tissues behave as the tumor tissue. Alternatively, if it is confirmed that CD74 overexpression has similar consequences in effector CD4+ and/or CD8+ T cells, CAR-T cells designed with the CD74 intracellular domain could increase T-cell infiltration and accumulation in the tumor. Thus, targeting CD74 represent a therapeutic target to positively or negatively modulate T-cell accumulation in the tissues, depending on the pathology.

## Methods

### Ethics statement

The experimental procedures using human samples are in line with the guideline of the Declaration of Helsinki and informed consents were obtained both from cancer patients and from HD. The protocol has been approved by the Ethic Committee of Institut Curie Hospital group (CRI- DATA190154).

Animal care and use for this study were performed in accordance with the recommendations of the European Community (2010/63/UE) for the care and use of laboratory animals. Experimental procedures were specifically approved by the ethics committee from Institut Curie, officially registered as CEEA-IC #118 and the Ministère de l'enseignement supérieur, de la recherche et de l'innovation which validated the project with the reference (APA-FIS#39042−2022103116127877 v2) in compliance with the international guidelines.

### Collection of clinical samples and cell isolation

Samples from patients with NSCLC are collected at the Institut Mutualiste Montsouris (Paris, France). Matched samples of blood, tumor-draining lymph node and tumor are collected from surgical residues available after a standard surgical resection followed by a histopathological analysis. Human samples are cut in small pieces and digested with Liberase TL (0.1 mg/ml, Roche) and DNase (0.1 mg/ml, Roche) for 30 min at 37 °C with agitation. The cell suspension is recovered in $CO_2$-independent medium (GIBCO) with 0.4% bovine serum albumin (BSA) and filtered in a 40 μm cell strainer (BD).

### Isolation of regulatory T cells

Blood buffy coats from HD are collected at the Etablissement Français du sang (Paris, France). PBMCs are obtained by gradient centrifugation using ficoll tubes filled with lymphoprep (STEMCELL - Cat#07851). CD25+ cells are isolated from total PBMCs by a positive selection with the CD25 Microbeads II kit (Miltenyi – Cat#130-092-983). Recovered cells are washed with PBS and stained with Live-Dead fixable Aqua, anti-CD4, anti-CD8, anti-CD25 and anti-CD127. Cells are resuspended at $20 \cdot 10^6$ cell/mL in PBS with EDTA (2 mM) and FBS (0.5%). Cells are FACS sorted as Aqua-negative, CD4-positive, CD127-negative and CD25high-positive. The sorted Tregs are centrifugated and resuspended at $10^6$ cell/mL in X-vivo medium completed with IL-2 (300 IU/mL, Novartis – Proleukine) and with soluble αCD3αCD28αCD2 (25 μL/mL, STEMCELL – Cat#10970) for genome edition or with αCD3αCD28 beads

(1bead:1cell, THERMOFISHER – Cat#11132D) for expansion. 10,000 Tregs are stained with anti-Foxp3 antibody after fixation and permeabilization to evaluate the expression of Foxp3 on the sorted Tregs before expansion. The X-vivo 15 media (OZYME – Cat#BE02-060F) is completed with human serum (10%), β-mercaptoethanol (50 μM), Pen/Strep (1%), non-essential amino acids (1%). To expand sorted Tregs, fresh IL-2 is added every 3 days and cells are reactivated every week. All the antibody characteristics are summarized in Table S1.

## Genome edition with CRISPR-Cas9 technique
Expanded sorted Tregs are re-activated with soluble αCD3αCD28αCD2 (25 μL/mL) 24 h before genome edition by electroporation. CRISPR RNA (crRNA, guides predesigned on IDT DNA) and trans-activating crRNAs (CAT#1072554 IDT DNA) are mixed at an equimolar ratio and incubate 30 min in a 37 °C-5% $CO_2$ incubator to form a single-guide RNA at 100 μM. CD74 guides: AltR1/rGrA rArGrG rUrGrU rArCrC rCrGrC rCrArC rUrGrA rGrUrU rUrUrA rGrArG rCrUrA rUrGrC rU/AltR2 and AltR1/rUrU rGrArG rArGrC rUrGrG rArUrG rCrArC rCrArU rGrUrU rUrUrA rGrArG rCrUrA rUrGrC rU/AltR2; and MIF guide: AltR1/rCrA rCrArG rCrArU rCrGrG rCrArA rGrArU rCrGrG rGrUrU rUrUrA rGrArG rCrUrA rUrGrC rU/AltR2. HiFi-Cas9 protein (IDT – Cat#1081060) is mixed with sgRNA at a ratio 1sgRNA:2Cas9 to form a CRISPR ribonucleoproteic complex (crRNP) at a concentration of 50 μM. Cell suspension is done at 1·10⁶cell in 20 μL of the supplemented *P3 Primary Cell* or *SE Cell Line* nucleofector solution (Lonza – for expanded Tregs Cat#V4XP-3032 and for MDA-MB231 cells Cat#V4XC-1024). 5 μL of crRNP (50 μM) is mixed to 20 μL of cell suspension and incubated at room temperature for 10 min. The cells with crRNP are transferred into a nucleofector cuvette and electroporate at EH100 pulse (Lonza – Cat#AAF-1003X). 5 days after the electroporation, the efficacy of the gene deletion is assessed by FACS and the electroporated Tregs are reactivated with IL-2 (300 IU/mL) and αCD3αCD28-coated beads (1bead:1cell) for further expansion.

## Retroviral transfection of human primary Tregs
Sorted Tregs are stimulated with IL-2 (300 IU/mL) and with αCD3αCD28-coated beads (1bead:1cell) 24 h before transduction. Tregs are transferred in retronectin coated plate and transduced by spinoculation with concentrated retrovirus. Retrovirus are designed with VectorBuilder with two promoters, the first one for the target sequence (or mCherry as a control) and the second one for EGFP. Seven days after spinoculation, transduction efficacy is assessed by FACS and Tregs are reactivated for further expansion.

## Suppressive assay
CD4+ cells are enriched from total PBMCs using the CD4 negative selection kit (BioLegend – Cat#76461). CD4+ cells are stained with cell-trace proliferation dye by incubating 30 min at 37 °C with CTV (1/1000). Tconvs are washed and the staining of CTV is evaluated by cytometry. 100,000 Tconvs are co-cultured with serial dilution of unstained Tregs from 1:1 to 1:32 Tregs:Tconvs. The co-cultured cells are in X-vivo completed media with αCD3αCD28 beads at the concentration 1bead:10cells. 5 days after, the proliferation peaks obtained by CTV dilution in the Tconvs are assessed by cytometry.

## Generation of tumor supernatant
The tumor-cell lines are plated in RPMI media completed with 10% FBS and 1% of penicillin/streptomycin for 48 h at 37 °C-5%$CO_2$. The supernatants are recovered and filtrated in a 0.22 μm filter and can be used directly or frozen and stored at –80 °C.

## Stimulation with cytokines or tumor-derived supernatant
Activation beads are removed from Tregs 24 h before stimulation. Tregs are resuspended at 100,000cell/100 μL of Xvivo media in a 96-well plate. 100 μL of Xvivo media containing the cytokine is added to the cells, beads and PMA-ionomycine are used as a positive control and

a well containing only Xvivo as a negative control. All the information for the stimulation is detailed in Table S2.

## Microchannels and micropillars for live imaging
Microfabricated chips for migration assays are prepared as previously described[5]. In brief, microchannels and micropillars are replicated with PDMS (RTV615, Neyco) from an epoxy mold. The PDMS chip is assembled on a glass bottom fluorodish (FD35-100, WPI) using plasma activation of surfaces. Microchannels used are 4 μm wide and 5 μm high and approximatively 900 μm long. Micropillars are composed of 10 μm-diameter cylinders spread at 30 μm distance. After mounting, the microchannels and micropillars are coated with Fibronectin (10 mg/mL, Sigma) following plasma activation of surfaces, then washed three times with PBS, then three times with Xvivo complete medium. A 2 mm-well is created at the entrance of the chips (microchannel and micropillar gels) to load the cells resuspended in Xvivo media. Then, DAPI-stained WT or CD74KO Tregs are loaded in the chip and let 2 h to enter the microchannel / micropillar area. Images are recorded in a 37 °C-5%$CO_2$ chamber with a frequency of one image every 3 min for 16 h, using an epifluorescence Nikon TiE microscope equipped with a cooled CCD camera (HQ2, Photometrics), using a 10X (NA = 0.3) dry objective. Image analysis is performed using home-made custom macros in Fiji software.

## Bulk-RNA sequencing of primary Tregs
1·10⁶ WT Tregs and 1·10⁶ CD74KO Tregs are washed in PBS 14 days after electroporation. Three biological replicates of each condition are used for RNA sequencing. Total RNA is isolated using RNeasy kit following the manufacturer's protocol (NORGEN – Cat#51800). After RNA quantification and quality assessment, retrotranscription is done to obtained cDNA with KAPA Hyper Plus Library (Roche). cDNA is amplified by PCR and cDNA library is sequenced with Illumina NovaSeq-S1-PE100 (35 Millions reads/sample). All the 6 samples are processed at the same time and on the same pipeline for sequencing. Reads are trimmed with TrimGalore using standard settings to remove adapters, sequenced poly-A tails and low-quality input. The trimmed reads are mapped on the human genome with the reference hg38 using STAR (version 2.7.1.a). Gene expression is quantified using featureCounts from the software package Subread. Downstream analysis is performed using R (version 4.1.1) in which raw counts are loaded and then normalized using R package DESeq2 (version 1.34.0). Statistical analysis to identify the differential expression genes is performed using DESeq2. Genes are considered significantly differentially expressed according to Benjamini−Hochberg adjusted *p*-value. Violin plots and volcanoplots are created using ggplot2 (version 3.3.6). Heatmaps are computed with R package Pheatmap (version 1.0.12). Enrichment pathways analyses are performed using enrichr function from EnrichR (version 3.0) and gseGO function from clusterProfiler (version 4.2.2). Enriched pathways for the two conditions are identified using the GeneOntolgy database (cellular component, molecular function, and biological process pathways).

## Antibody staining and flow cytometry
Cells are stained with a viability marker in PBS during 15 min at 4 °C. Following this step, they are stained with a mix of fluorochrome-conjugated antibodies against surface protein in staining buffer (PBS, 2 mM EDTA, 0.5% BSA) during 20 min at 4 °C. After two washes with staining buffer, cells are fixed in Fixation/Permeabilization kit during 30 min at 4 °C (ThermoFisher scientific−Cat#15151976) and washed twice with Permeabilization Buffer (ThermoFisher – Cat# 12766048). Intracellular staining is done by diluting the fluorochrome -conjugated antibodies in the Permeabilization Buffer during 20 min at room temperature protected from light. Cells are washed twice with staining buffer and analyzed on a Fortessa flowcytometer (BD) or ZE5 flow-cytometer (BioRad) then using FlowJo software (version 10). All the antibody characteristics are summarized in Table S1.

## Cytokine staining

Cells are incubated 3 h at 37 °C with Xvivo with or without stimulation, Golgi Stop and Golgi Plug are added for 1 more hour of incubation at 37 °C. Cells are washed with PBS. Cells are stained for viability and surface marker, then cells are fixed and permeabilized as described in "Antibody staining and flow cytometry" section. Anti-Human cytokine antibodies are then added in the mix of intracellular antibodies. All the antibody characteristics are summarized in Table S1, and all the information for the stimulation are detailed in Table S2.

## Methylation status of Foxp3 Treg-specific demethylated region (TSDR)

Expanded Tregs and total CD4+ cells from paired HD are washed twice in PBS. Genomic DNA is treated with Bisulfite according to manufacture indication (EZ DNA Methylation-Direct Kit Zymo Research #D5020). Then, a quantitative PCR is performed for specific methylation and demethylation of Foxp3 TSDR locus in a 10 μL reaction with 25 ng of bisulfite-treated DNA, sybrGreen and 0.5 μM of forward and reverse primers. FOXP3-TSDR demethylation-specific primers, forward: 5′-TAGGGTAGTTAGTTTTTGGAATGA-3′, reverse: 5′-CCATTAACATCATAACAACCAAA-3′. FOXP3-TSDR methylation-specific primers, forward: 5′-CGATAGGGTAGTTAGTTTTCGGAAC-3′, reverse: 5′-CATTAACGTCATAACGACCGAA-3′[41]. Finally, the percentage of Foxp3 TSDR demethylation is calculated as $100/(1 + 2^{[Ct(methylation)-Ct(demethylation)]})$.

## Immunostaining

Cells are washed in PBS and seeded on L-polylysine coated slides 30 min at 37 °C (300,000cell/slide). After one wash with PBS, cells are fixed in PFA for 10 min at room temperature. The PFA is removed and free PFA is quenched by glycine (10 mM) for 10 min at rT. The cells are then permeabilized with saponin and stained with the primary antibodies for 1 h at rT. After wash, the cells are incubated with the secondary antibody for 30 min at rT. All the antibody characteristics are summarized in Table S1. Image acquisition is done in a 4-laser confocal microscope Leica SP8.

## Electronic microscopy

After 24 h of rest, Tregs are seeded on concanavalin A–coated coverslips (Sigma) and incubated for 30 min at 37 °C. Cells are then fixed in 2,5% glutaraldehyde in 0.1 M cacodylate buffer, pH 7.4 for 1 h, post-fixed for 1 h with 2% buffered osmium tetroxide, dehydrated in a graded series of ethanol solution, and then embedded in epoxy resin. Images were acquired with a digital camera Quemesa (SIS) mounted on a Tecnai Spirit transmission electron microscope (FEI Company) operated at 80 kV.

## Tumor cell lines

B16-F10 (CRL-6475) murine melanoma and HCC827 (CRL-2868) human lung tumor cell lines were obtained from the ATCC. MCA101 fibrosarcoma was kindly given by Clotilde Thery and MDA-MB231, MCF-7 by Pascale Hubert (Institut Curie, Paris, France), MC38 by Nicole Haynes (Peter MacCallum Cancer Centre, Melbourne, Australia). Cells were cultured at 37 °C-5%CO₂ in RPMI 1640 (Gibco) supplemented with 10% FBS (Eurobio). At 80% of confluence, cells were detached with trypsin-EDTA (Gibco) and rinsed twice in PBS. Cell lines were checked for the absence of *Mycoplasma*.

## Mice

C57Bl/6 were purchased from Charles River (France) and used after at least 1 week of adaptation. NSG mice (NOD.Cg-*Prkdc^scid Il2rg^tm1Wjl*/SzJ, JAX#005557) were bred at CNRS Central Animal Facility TAAM (Orléans, France). Female CD45.1 C57Bl/6, Bl/6 Rag2⁻/⁻ KP as a source of Rag2⁻/⁻mice[42], and CD45.2 *Cd74⁻/⁻* C57Bl/6 mice (CD74KO, kindly provided by A.M. Lennon, Institut Curie, France) were maintained and handled at Curie Institute facility. Female mice used in the experiments were age-matched and were euthanized by cervical dislocation. Mice

breeding was in SPF animal facilities and experimental and control animals were co-housed with housing conditions using a 12 light/12 dark cycle, with a temperature between 20 and 24 °C with an average humidity rate between 40% and 70%. Human endpoints were used for tumor-bearing mice as maximal ethical size of tumors subcutaneously grafted of 2 cm³ (calculated as (length × width²)/2), more than 20% of weight loss, signs of altered mobility-eating ability and cachexia.

## Generation of *Cd74^fl/fl* knock-in mice

The mouse *Cd74* gene (ENSMUSG00000024610) was edited using a double-stranded homology-directed repair template (targeting vector) containing 1300 and 800 bp-long 5′ and 3′ homology arms, respectively. It included a loxP site and a frt-neor-frt cassette that were both inserted in intron 1, 83 bp upstream of the 5′ end of exon 2, and a loxP site located in intron 3, 87 bp downstream of the 3′ end of exon 3. The final targeting vector was abutted to a cassette coding for the diphtheria toxin fragment A[43]. Two sgRNA-containing pX330 plasmids (pSpCas9; Addgene, plasmid ID 42230) were constructed. In the first plasmid, two sgRNA-specifying oligonucleotide sequences (5′-CACCGCTCAAACCCTCCCAATAGTG-3′ and 5′-AAACCACTATTGGGAGGGTTTGAGC-3′) were annealed, generating overhangs for ligation into the BbsI site of plasmid pX330. In the second plasmid, two sgRNA-specifying oligonucleotide sequences (5′-CACCGGAATCTGGGCACTAACTAC-3′ and 5′-AAACGTAGTTAGTGCCCAGATTCC-3′) were annealed and cloned into the BbsI site of plasmid pX330. The protospacer-adjacent motifs (PAM) corresponding to each sgRNA and present in the targeting vectors were destroyed via silent mutations to prevent CRISPR-Cas9 cleavage. JM8.F6 C57BL/6N ES cells[44] were electroporated with 20 μg of targeting vector and 2.5 μg of each sgRNA-containing pX330 plasmids. After selection in G418, ES cell clones were screened for proper homologous recombination by Southern blot and PCR analysis. A neomycin specific probe was used to ensure that adventitious non-homologous recombination events had not occurred in the selected ES clones. Mutant ES cells were injected into BALB/cAnNRj blastocysts. Following germline transmission, excision of the frt-neor-frt cassette was achieved through genetic cross with transgenic mice expressing a FLP recombinase under the control of the actin promoter[45].

A single pair of primers (sense 5′-AACATGCTCCTTGGGGTAAGG-3′ and antisense 5′-TTGCAAATTGTGGGACTTGCC-3′) was used to distinguish the WT and *Cd74^fl* mutant allele, amplifying a 218 bp-long and a 303 bp-long band, respectively. The resulting *Cd74^fl* mice (official name C57BL/6NRj-Cd74^tmICiphe mice) have been established on a C57BL/6NRj background. When bred to mice that express tissue-specific Cre recombinase, the resulting offspring will have exons 2 and 3 removed in the Cre-expressing tissues, resulting in cells lacking CD74 expression.

## Generation of Foxp3-specific CD74 conditional knockout mice (CD74cKO mice)

Homozygous CD74^flox mice (CD74^fl/fl) mice were crossed with Foxp3^eGFP-Cre-ERT2 mice (JAX#016961)[46], kindly given by Dr. J. Kanellopoulos (Université Paris-Saclay, France).

## Tamoxifen treatment

Tamoxifen (Sigma-Aldrich) was resuspended in peanut oil (Sigma-Aldrich) at a final concentration of 40 mg/ml and 8 mg was administered once a week by oral gavage on days 0, 7 and 14.

## Generation of Graft-versus-Host disease model in immunodeficient host

After 7 days of adaptation, NSG mice are injected intravenously with 100 μL of PBS containing the different cell mixes. Total PBMC are stained to evaluate the percentage of CD3+ cells. The three different cell suspensions are made of 5.10⁶ CD3+ cells/50 μL with 1−50 μL of PBS; 2−50 μL containing 5.10⁶ WT Tregs; 3−5.10⁶ CD74KO Tregs. Mice are assessed twice a week for the GvHD signs: weight loss, skin

integrity, posture, fur texture, activity and diarrhea. For survival study, mice are monitored 5 times a week and unhealthy mice, according to ethical grade, are euthanized and counted as non-survivors. 3 mice from each group are euthanized 14 days after cell injection, and liver and spleen are collected for cytometry analysis. Livers and spleens are processed to extract the cell suspension and a staining is performed to analyze the human cells. For the liver, a percoll is done to remove debris.

### Generation of tumor model in humanized mice

After 7 days of adaptation, NSG mice are subcutaneously injected on the flank with 100 μL of PBS containing $5.10^6$ MDA-MB231 breast tumor cells (day 0). When the tumor is palpable (between day 6 and day 10), mice are injected intravenously with 100 μL of PBS containing the different cell mixes. The preparation of the mixes is similar to the GvHD model. Mice are assessed twice a week to measure the tumor size. When the tumor exceeds the ethical size, mice are euthanized. 3 mice from each group are euthanized 6 days after cell injection and liver, spleen and tumor are collected for cytometry analysis. Livers and spleens are processed to extract cells. For liver and tumor, a percoll is done to remove debris. Cells from spleens, livers and tumors are stained to evaluate the phenotype and repartition of Tregs. For the experiment with MIFKO tumor, a similar method is followed. Briefly, using CRISPRCas9 technology, we generated MDA-MB231 cells deficient for MIF, as described previously in "Genome edition with CRISPR-Cas9 technique", and obtained MIF+ (control) or MIFKO MDA-MB231 tumor cells. MIF+ or MIFKO MDA-MB231 tumor cells are engrafted s.c. and 10 days later, PBMCs, WT and CD74KO expTregs are injected intravenously. At day 16 after tumor engraftment, the spleen, liver, and tumor are harvested by FACS to evaluate WT and CD74KO expTreg repartition.

### Anti-CD74 treatment in humanized mice

As described previously in tumor model in humanized mice section, NSG mice are subcutaneously injected on the flank with 100 μL of PBS containing $5.10^6$ MDA-MB231 cells (day 0) and 14 days later, a mix of human T cells, PBMCs + WT expTregs, is transferred intravenously. On day 14, 2 h before T-cell transfer, mice are left untreated, treated with anti-CD74 antibody (Milatuzumab Recombinant Human antibody, ThermoFisher Cat#MA5-41757) by an intravenous (i.v.) or intratumor (i.t.) injection. Anti-CD74 is administrated in at 100 μg in 100 μL of PBS i.v. and 10 μg in 50 μL of PBS i.t. The anti-CD74 treatment is repeated on day 16 and 18, and on day 20, spleens, livers, and tumors are recovered to evaluate PBMC-derived CD4 and expTreg infiltration into the tissues.

### Tumor models in syngeneic mice

Mouse tumor cell lines are grafted subcutaneously in C57Bl/6 mice at $0.5 × 10^6$ cell/mouse. Tumor growth and body weight are monitored to verify the effect of tumor induction in mouse wellbeing. Mice were randomly assigned a treatment group and tumor volume determined by caliper measurements. On day 21 after transplant, spleen, tumor-draining lymph nodes and tumors are collected. Tissues are processed by mechanic dissection follow by enzymatic digestion only for the tumor. For liver and tumor, a percoll is performed to enrich in lymphocytes. Cell suspensions from the different tissues are then stained with fluorochrome-coupled antibodies for FACS analysis.

### Adoptive transfer of CD4+CD25+ cells in syngeneic tumor model

Rag2$^{-/-}$ KP mice[42] are subcutaneously injected on the flank with 100 μL of PBS containing $0.5 × 10^6$ MCA101 tumor cells. On day 12, mice receive a mix of CD45.2+CD74KO CD4+CD25+ splenocytes (obtained from CD74 full KO mice) and CD45.1+CD4+CD25+ splenocytes (obtained from WT littermates). CD4+CD25+ splenocytes were obtained by successive enrichment using magnetic columns (negative selection of CD4+ T cells (MojoSort, Biolegend #480005)

followed by CD25 positive selection (MicroBead, Miltenyi #130-091-072), and a mix of $1×10^6$ cells are injected in 100 μL by i.v. On day 18 (6 days after cell transfer), spleens and tumors are collected for cytometry analysis. The injected mix of CD45.2 CD74KO/CD45.1 WT CD4+CD25+ splenocytes contained a proportion of 84/16 for replicate-1 (n = 1 mouse) and of 71/29 for replicate-2 (n = 2 mice). To take into consideration the variability of the CD45.2/CD45.1 ratio in the injected mix, results are expressed as the CD45.2/CD45.1 ratio in the tissue, normalized to the initial CD45.2/CD45.1 ratio.

### Reporting summary

Further information on research design is available in the Nature Portfolio Reporting Summary linked to this article.

## Data availability

In-house bulk-RNAseq has been deposited in GEO database with the GEO accession number: GSE229389. Tosello, Richer et al. dataset (NSCLC) has been deposited in EGA, with accession code EGAS50000000293 for scRNAseq and EGAS50000000294 for scATAC-seq. The sequence data are generated from patient samples and therefore are available under restricted access. Data access can be granted via the EGA with completion of an institute data transfer agreement, and data will be available for one year once access has been granted. For De Simone et al. dataset, the original GEO accession numbers are GSE40419 (NSCLC) and GSE50760 (CRC). For Nunez et al. dataset, RNAseq data are available in ArrayExpress with accession code E-MTAB-9112. The remaining data are available within the Article, Supplementary Information or Source Data file. Source data are provided with this paper.

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

## Acknowledgements

We acknowledge all the U932 members for the helpful discussion, Yago A. Arribas and Blandine Baudon for the technical support; as well as Egle-Tx employees, Mercedes Tkach and Irena Chacon for experiment advice. In addition, we thank the AVIESAN – IFB Bioinformatic school for the training to analyze Bulk RNA-seq data, Adrien Pain for the tutorial. We thank Lea Guyonnet, Anna Chipont, Annick Viguier, and Coralie Guerin from the Curiecoretech Cytometry platform at Institut Curie; and the Curiecoretech Animal facility platform and Next Generation Sequencing platform at Institut Curie. The *Cd74*fl knock-in mice were developed by Centre d'Immunophénomique (CIPHE). CIPHE is supported by the Investissement d'Avenir program PHENOMIN (French National Infrastructure for mouse Phenogenomics; ANR10-INBS-07) and Agence Nationale de La Recherche (LABEX DCBIOL, ANR-10-IDEX-0001-02 PSL, and ANR-11-LABX-0043). This work (E.B., M.R.R., F.M., R.M.O., J.D., M.M., J.J.S., M.J., P.E.B., W.R., N.G., E.d.B., H.D.M., C.S., C.H., A.M.L., J.T.B. and E.P.) has received the support of the LabEx DCBIOL (ANR-10-IDEX-0001-02 PSL; ANR-11-LABX-0043); INCa-DGOS-Inserm_12554, and Center of Clinical Investigation (CIC IGR- Curie 1428) and Egle-therapeutics (F.N., S.L.).

## Author contributions

Conceptualization and methodology: E.l.B., C.S., J.T.B., E.P. Experiment: E.l.B., J.D., C.S., J.T.B, M.R.R., F.M., R.M.O. Microscopy imaging: E.l.B., M.J., J.J.S., H.D.M.. Transduction: E.l.B., F.N.. Data analysis: E.l.B., M.M., J.T.B., P.E.B. Data management and R-language support: P.E.B, W.R., SL.

NGS support: S.B. Clinical samples: N.G., M.L., E.d.B., A.V.S.. Investiation and discussion: A.L.D., C.H. Supervision: J.T.B., E.P. Writing of original draft: E.l.B., C.S., A.L.D., J.T.B., E.P.

## Competing interests

E.P. is co-founder and consultant for Egle Therapeutics and is shareholders at Mnemo Therapeutics. F.N. and S.L. are employees of Egle Therapeutics. J.T.B. is consultant for Egle Therapeutics. P.E.B. is consultant for Mnemo Therapeutics. The remaining authors declare no competing interests.
