## [Peer Review File · Nature Communications]

CD74 supports accumulation and function of regulatory T cells in tumorsREVIEWER COMMENTS

Reviewer #1 (Remarks to the Author): with expertise in Tregs, cancer immunology

Bonin et al report their observations on the potential role of CD74 (HLA-II invariant chain) in tumor infiltrating regulatory T cells. The work is motivated by the search for Treg selective therapeutic targets in cancers. Based on public datasets the authors identify CD74 as a differentially upregulated molecule on the surface of human tumor infiltrating Treg as compared to Tconv or to T cells at other sites. The authors generated CD74k.o. and CD74 overexpressing variants of human Treg for extensive phenotyping and functional analyses both in vitro and, in xenotransplant tumor mouse models, in vivo. They report in CD74 deficient cells a reduction of HLADR expression, reduced accumulation/maintenance in tumor tissue, loss of tolerizing function and instability of FoxP3 expression in vivo and altered/impaired migratory capacity associated with transcriptomic and morphological changes in cell membranes, organelle structure and cytoskeletal architecture. They conclude that particularly under inflammatory conditions such as in the tumor microenvironment CD74 is necessary to stabilize a suppressive phenotype and maintain Treg infiltration high in tumors. Thus CD74 represents an interesting candidate for therapeutic intervention in cancer immunotherapy.

The authors present a substantial body of experimental data that support the overall conclusion of a functional impact of CD74 in Treg biology, particularly within tumors. The experiments are in general well designed and conducted, the manuscript is very well written and the presented data appear largely suggestive. Still, there is a major concern in drawing fundamental functional conclusions on the general biology of a human T cell subset from studies in a xenograft environment, particularly in regard to T cell migration, accumulation and maintenance. Many parameters of the lymphatic system, the vasculature and the complex tumor microenvironment influence the fitness, stability and functional behaviour of Treg cells and T_H17 cells in situ and most of these are not matched in xenograft systems. Therefore, The disclosed data and key conclusions are not sufficiently validated as long as they are not accompanied by respective experiments in a syngeneic mouse model. In addition, there is already a considerable number of studies that investigated a large diversity of biological functions of CD74 in various cells of the immune system, including T and B cells, apart from its core role as a chaperone in HLA-II loading. Moreover, molecular

binding partners, signalling circuits and transcriptional programs related to CD74 activation have been described that roughly match those characterized by the authors for Treg. While these observations were not specifically reported on Treg, they somewhat dampen the fundamental character of the findings described here. In order to provide such substantial step forward, the authors could dissect the cause for the supposed different expression of CD74 in Tregs versus Tconv/CD8TC, clarify whether and why CD74 mediates Treg immigration or alternatively Treg maintenance in the tumor or how FoxP3 stabilization is affected on a molecular level. Here, the authors fail to embark on previous studies demonstrating e.g. NFκB activation through CD74 signalling, or exploring the role of MIF or other natural CD74 ligands and their abundance in the tumor microenvironment. Still, the reported findings are of particular translational interest and will attract broader attention of researchers in the field of cancer immunology and immunotherapy.

Reviewer #2 (Remarks to the Author): with expertise in CD74

The paper by E. Bonnin et al. reveals yet another function of the CD74 molecule in a very special cell type, the regulatory T-cells.

They show in a xenograft mouse model that human Treg that reside in human breast carcinoma transplants inhibit the GVHD-induced tumor rejection brought about by coinoculating these mice with human PBMCs.

They further show by a CRISPER-Cas9 knock out of CD74 in human Treg that the inhibitory effect of these manipulated cells on GVHD-induced breast carcinoma rejection is impeded, allowing the conclusion that CD74 is causative. This assumption is tested by forced hyperexpression of CD74 in Treg and the result supports this concept in that these CD74 hyperexpressing cells exert an even stronger inhibitory effect on tumor rejection.

They proceed in explaining this effect by a change in migratory capacity of these CD74KO Treg which they show by ex vivo migration assays are unable to properly migrate.

The last set of experiments focusses the intracellular changes in Treg when CD74 is knocked down. Forced CD74 deficiency leads to structural alterations, an hairy phenotype, cytoskeletal changes and altered gene expression in a number of genes.

The paper is perfectly written and the data as they are presented in a large number of

complex Figures are convincing as such.

However, I am especially surprised that the different phenotypes of isolated and genetically manipulated Treg seem to be stable over time *ex vivo* and that these phenotypes seem to be rather traits than different states of activation since these isolated and manipulated subsets of Treg can even be expanded by repetitive IL-2 treatment. This further implies that activation induced death does not occur.

In my opinion, this complex study is a high-wire performance in highly artificial systems. Looking at it from far below as a grounded pathologist who performs PD-1 immunohistochemistry every day to detect and quantify intratumoral T-cells expressing PD-1, this study lacks the connection to the ground, i.e., to the real life situation in cancer patients. Most carcinomas and melanomas contain very few tumor infiltrating T-cells and even less Treg.

Reviewer #3 (Remarks to the Author): with expertise in Tregs, cancer immunology

The authors demonstrate that Tregs express surface CD74 in a tumor microenvironment-dependent manner. CD74 specifically stabilizes the suppressive phenotype of Tregs within tumors and is required for the migration and retention of intratumoral Tregs. The authors provide evidence that Treg retention in tumor may reflect CD74 triggered reorganization of Tregs' intracellular organelles and capacity to interact with neighboring cells.

As the invariant chain of MHC-II, CD74 was initially thought to be mainly restricted to antigen presenting cells (APCs). The authors report for the first time that CD74 is exclusively expressed by tumor infiltrated Tregs and specifically stabilizes their suppressive phenotype. These findings represent a potentially important conceptual advance in our understanding of Treg biology in the tumor microenvironment and also provide insight into developing therapeutic approaches targeting CD74. However, there are critical issues that need to be addressed before consideration for publication.

Major critique:

1. The authors claim that CD74 specifically regulates the Treg suppressive phenotype in tumor according to the observation that CD74KO Tregs are exclusively dysfunctional in the tumor tissue, but not in vivo in a GVHD model. However, the suppressive activity of CD74KO Tregs in GVHD may not be an appropriate model for comparison, since the allo-reactive response in the GVHD model lacks or masks the tumor cues that may upregulate Treg surface CD74 expression. A direct comparison between intratumoral Treg and in non-tumor tissues from Treg CD74 expression and the Treg phenotype would be more informative.

2. A portion (6-22%) of intratumoral Treg express CD74, while 80%-99% of conventional T cells and Tregs express intracellular CD74. Is there a functional difference between surface CD74 and intracellular CD74? It is critical to distinguish the function of different forms of CD74 in intratumoral Treg using anti CD74 that blocks surface CD74 and spares intracellular forms. This will also rule out the potential impact of intracellular CD74.

3. “The enhanced tumor rejection observed with the transfer of CD74KO expTregs suggests that in the absence of CD74, Tregs no longer suppress antitumor immune responses but may further acquire anti-tumoral functions.” Does this mean KO CD74 converts Treg into effector-like T cells? Since CD74 deletion does not affect in vitro or in vivo expression of key molecules, the authors should characterize the phenotype of CD74 KO Treg in this model in terms of acquisition, e.g., acquisition of effector cytokine expression (e.g., IFN γ) or reduction of suppressive cytokines (e.g., IL-10, TGF- β).

4. The authors indicate that CD74 may act as a receptor for chemokines produced by the tumor, which accounts for the retention of Tregs within tumors. Perhaps the authors could discuss potential mechanisms, e.g, chemokine-CD74 interaction that might underlie Treg migration.

5. Bulk RNA-seq analysis profiling WT and CD74KO Tregs was used to define the molecular mechanisms underlying the effect of CD74 on Treg motility and tumor retention. The authors should validate DEG pathway gene products at the protein level to validate these conclusions.

6. For the co-culture experiment in Fig.S7, it's not clear how Treg contacting neighboring cells by exchange mitochondria contributes to Treg stability and retention in the tumor. Although mesenchymal stem cells may transfer mitochondria to Treg to enhance their immunosuppressive function, it may not apply to the PBMC-Treg contacting. Moreover, the co-culture assay was performed in a tumor-free setting, which cannot mirror the environmental cues in the tumors.

Minor points:

1. Introduction: Reference #1 is a submitted manuscript, which is not appropriate as cited literature. The authors should state that the relevant findings are in a manuscript currently under consideration.

2. Fig. S5B-D: A CD74KO group should also be included in addition to CD74OE group. It will be more convincing to compare the impact of CD74KO and CD74OE on Treg tumor accumulation side-by-side.

Editorial Note: Cartoon schematics were created with BioRender.com.

REVIEWER COMMENTS

Reviewer #1 (Remarks to the Author): with expertise in Tregs, cancer immunology

Reviewer #1: Bonnin et al report their observations on the potential role of CD74 (HLA-II invariant chain) in tumor infiltrating regulatory T cells. The work is motivated by the search for Treg selective therapeutic targets in cancers. Based on public datasets the authors identify CD74 as a differentially upregulated molecule on the surface of human tumor infiltrating Treg as compared to Tconv or to T cells at other sites. The authors generated CD74k.o. and CD74 overexpressing variants of human Treg for extensive phenotyping and functional analyses both in vitro and, in xenotransplant tumor mouse models, in vivo. They report in CD74 deficient cells a reduction of HLADR expression, reduced accumulation/maintenance in tumor tissue, loss of tolerizing function and instability of FoxP3 expression in vivo and altered/impaired migratory capacity associated with transcriptomic and morphological changes in cell membranes, organelle structure and cytoskeletal architecture. They conclude that particularly under inflammatory conditions such as in the tumor microenvironment CD74 is necessary to stabilize a suppressive phenotype and maintain Treg infiltration high in tumors. Thus CD74 represents an interesting candidate for therapeutic intervention in cancer immunotherapy.

The authors present a substantial body of experimental data that support the overall conclusion of a functional impact of CD74 in Treg biology, particularly within tumors. The experiments are in general well designed and conducted, the manuscript is very well written and the presented data appear largely suggestive. Still, there is a major concern in drawing fundamental functional conclusions on the general biology of a human T cell subset from studies in a xenograft environment, particularly in regard to T cell migration, accumulation and maintenance. Many parameters of the lymphatic system, the vasculature and the complex tumor microenvironment influence the fitness, stability and functional behaviour of Treg cells and T_H17 cells in situ and most of these are not matched in xenograft systems. Therefore, The disclosed data and key conclusions are not sufficiently validated as long as they are not accompanied by respective experiments in a syngeneic mouse model.

In response to the reviewer's valuable input, we have undertaken additional experiments using three distinct syngeneic mouse models to address these concerns, including a new mouse model of inducible deletion of CD74 selectively on Tregs, that we have generated for this project. The new data obtained from syngeneic models align with and corroborate our observations made in human Tregs, providing a more robust validation of the functional impact of CD74 in Treg biology within the tumor microenvironment. We believe these additional experiments enhance the validity of our findings and provide a more comprehensive understanding of the role of CD74 in Treg biology across different model systems. We thank the reviewer for their insightful comments, which have contributed to the refinement and strengthening of our study.

#1 Validation of the initial observation: CD74 is overexpressed by tumor-Tregs in immunocompetent, WT mice.

Three weeks after the injection of B16-F10 melanoma cells subcutaneously in mice, we recovered the spleen, tumor-draining lymph nodes and tumor to evaluate the expression of CD74 in T cells (**Reviewer Figure 1**). We observed that, as in humans, the surface expression of CD74 was higher in tumor-Tregs compared to peripheral Tregs, and to tumor and peripheral CD4+ Tconvs and CD8+ T cells. Additionally, the vast majority of Tregs expressed CD74 intracellularly, although not at 100%.

These results (**Reviewer Figure 1 A and B**) are now included as **Figure 7A and B** in the revised version of the manuscript.

#2 Validation of the obtained result: CD74-KO Tregs infiltrate less the tumor compared with WT Tregs (experiments using CD74 full KO mice/cells).

In a similar approach to the human study, to characterize the role of CD74 in murine tumor Tregs, we engrafted B16-F10 tumors in CD74 full KO mice (CD74^{-/-}) or WT littermates and analyzed Treg infiltration in different tissues (**Reviewer Figure 2**). We observed that the frequency of Tregs among total CD4+ T cells was higher in spleens and tumor-draining lymph nodes of CD74^{-/-} mice compared to WT littermates; while in the tumor, the proportion of Tregs was similar in both mice, as if the CD74KO Tregs lost their enhanced capacity to infiltrate tissues only in the tumor. The limitation of this full CD74KO model is that all cells lack CD74 expression, including B cells and thymic APCs, and consequently, CD74KO mice present impaired thymic selection, which could highly impact on the biology of the T cells. Consequently, these results are difficult to interpret as the model does not allow distinguishing a cell-intrinsic or cell-extrinsic effect of CD74 on Treg behavior. Thus, we have decided not to include these results in the revised version of the manuscript, but we show them here for reviewer perusal.

Reviewer Figure 2: CD74-deficient mice display a decreased proportion of tumor Tregs compared to their WT counterpart. **A:** WT (black, n=7) or full CD74^{-/-} (red, n=6) C57BL/6 mice were engrafted with $0,5 \times 10^6$ B16-F10 melanoma cells, and 20 days later, the spleen, TDLN and tumor were analyzed by FACS. Representative dotplots of CD4⁺ T cells among live TCR β ⁺ cells. **B:** Ratio of the percentage of Tregs in CD74^{-/-} over the mean of the percentage of Tregs in WT mice. Statistical analyses are performed using an unpaired t-test with pVal < *:0,01; **:0,001; ***:0,0001.

Additionally, we designed a model of adoptive cell transfer to evaluate the capacity of CD74KO and WT Treg to infiltrate the tumor in the same host mice (**Reviewer Figure 3**). For this, we co-injected into MCA tumor-bearing Rag-KO mice Tregs (CD4⁺ CD25^{high}) sorted from CD45.1 WT or CD45.2 CD74^{-/-} mice at 1:1 ratio. After 6 days, we evaluated the infiltration of Tregs in the spleens, tumor-draining lymph nodes and tumors by FACS. We observed that the KO/WT Treg ratio was slightly lower in the tumor compared to the spleen. These results suggesting that CD74^{-/-} Tregs tend to accumulate less than the WT Tregs in the tumor compared to the spleen, point to a potential role for CD74 in tissue-accumulation and are in agreement with the previous observations in mice and men. However, as CD74KO cells were obtained from CD74 full KO animals, where the repertoire of T cells could be biased by the absence of CD74 in thymic APCs and epithelial cells, these results should be the interpreted with precaution.

Reviewer Figure 3: CD74-deficient Tregs display impaired infiltration into tumors. RagKO mice (n=3) were engrafted with $0,5 \times 10^6$ MCA cells, and 12 days later, a mix of CD4⁺CD25⁺ enriched splenocytes composed half of splenocytes from WT CD45.1 mice and half from CD45.2 CD74^{-/-} mice are transferred intravenously. Six days later, spleens and tumors were analyzed by FACS. Representative dotplots of Tregs among live TCR β ⁺ CD4⁺ T cells (**A**) and quantification of the ratio of CD45.2+ Tregs over CD45.1+ Tregs (**B**).

These results (**Reviewer Figure 3A and B**) are now included as **Figure 7C and 7D**, respectively, in the revised version of the manuscript.

To circumvent the cell-extrinsic effects of CD74KO mice models, **we generated Foxp3-specific CD74 conditional knock-out mice (CD74^{ckO} mice)**.

Below we added the section we have incorporated in Material and Methods for the description of the new mouse strains.

Generation of *Cd74*^{flox} knock-in mice

The mouse *Cd74* gene (ENSMUSG00000024610) was edited using a double-stranded homology-directed repair template (targeting vector) containing 1300 and 800 bp-long 5' and 3' homology arms, respectively. It included a loxP site and a frt-neor-frt cassette that were both inserted in intron 1, 83 bp upstream of the 5' end of exon 2, and a loxP site located in intron 3, 87 bp downstream of the 3' end of exon 3. The final targeting vector was abutted to a cassette coding for the diphtheria toxin fragment A (Soriano et al., 1997). Two sgRNA-containing pX330 plasmids (pSpCas9; Addgene, plasmid ID 42230) were constructed. In the first plasmid, two sgRNA specifying oligonucleotide sequences (5'-CACCGCTCAAACCCTCCCAATAGTG-3' and 5'-AAACCACTATTGGGAGGGTTTGAGC-3') were annealed, generating overhangs for ligation into the BbsI site of plasmid pX330. In the second plasmid, two sgRNA-specifying oligonucleotide sequences (5'-CACCGGAATCTGGGCACTAACTAC-3' and 5'-AAACGTAGTTAGTGGCCAGATTCC-3') were annealed and cloned into the BbsI site of plasmid pX330. The protospacer-adjacent motifs (PAM) corresponding to each sgRNA and present in the targeting vectors were destroyed via silent mutations to prevent CRISPR-Cas9 cleavage. JM8.F6 C57BL/6N ES cells (Pettitt et al., 2009) were electroporated with 20 µg of targeting vector and 2.5 µg of each sgRNA-containing pX330 plasmids. After selection in G418, ES cell clones were screened for proper homologous recombination by Southern blot and PCR analysis. A neomycin specific probe was used to ensure that adventitious non-homologous recombination events had not occurred in the selected ES clones. Mutant ES cells were injected into BALB/cAnNRj blastocysts. Following germline transmission, excision of the frt-neor-frt cassette was achieved through genetic cross with transgenic mice expressing a FLP recombinase under the control of the actin promoter (Rodriguez et al., 2000).

A single pair of primers (sense 5'-AACATGCTCCTTGGGGTAAGG-3' and antisense 5'-TTGCAAATTGTGGGACTTGCC -3') was used to distinguish the WT and *Cd74*^{fl} mutant allele, amplifying a 218 bp-long and a 303 bp-long band, respectively. The resulting *Cd74*^{fl} mice (official name C57BL/6NRj-*Cd74*^{tm1Ciphe} mice) have been established on a C57BL/6NRj background. When bred to mice that express tissue-specific Cre recombinase, the resulting offspring will have exons 2 and 3 removed in the Cre-expressing tissues, resulting in cells lacking CD74 expression.

Generation of Foxp3-specific CD74 conditional knock-out mice (CD74^{ckO} mice).

Homozygous CD74^{flox} mice (CD74^{fl/fl}) mice were crossed with Foxp3^{eGFP-Cre-ERT2} mice (JAX#016961) (Rubtsov et al., 2008), kindly given by Dr J. Kanellopoulos (Université Paris-Saclay, France).

Tamoxifen treatment

Tamoxifen (Sigma-Aldrich) was resuspended in peanut oil (Sigma-Aldrich) at a final concentration of 40mg/ml and 8mg was administered once a week by oral gavage on days -7, 0 and +7.

Results

Characterization of newly generated Foxp3-specific CD74 conditional knock-out mice (CD74^{ckO} mice).

The newly generated CD74^{flox} mice were validated by PCR (**Reviewer Figure 4A**). Results illustrate the Wt versus mutant alleles in heterozygous (CD74^{wt/fl}) or homozygous (CD74^{fl/fl}) mice generated by the insertion of loxP sequences. Subsequently, CD74^{fl/fl} mice were crossed with Foxp3^{eGFP-Cre-ERT2} mice. The resulting CD74^{ckO} mice were born in typical Mendelian proportions and developed normally. For the analysis of the Treg compartment, we measured GFP expression in CD4+CD25+ cells in non-fixed blood cells from the resulting offspring harboring the various possible genotypes for the ^{GFP-Cre-ERT2} modification (**Reviewer Figure 4B,C**). As expected, GFP was expressed in CD4+CD25+ cells of the homozygous Cre⁺/Cre⁺ females and Cre^{+/Y} males, and at lower levels in the heterozygous Cre⁺ mice. To validate the Treg-specific CD74 deletion, CD74^{ckO} mice were treated by tamoxifen and 4 days after the last injection (**Reviewer Figure 4D**), we quantified total (surface and intracellular) CD74 expression in splenic T cells. As observed in **Reviewer Figure 4E,F**, the deletion of CD74 upon Cre recombination induced by tamoxifen, was specific in the Tregs of the expected mice genotypes, with no apparent changes in Tconv and CD8+T cells. Notably, CD74 was not efficiently deleted in heterozygous females Cre^{-/+} mice. Consequently, in all subsequent experiments we only used male mice with CD74^{fl/fl} Foxp3^{eGFP-Cre-ERT2(+/Y)} (hereafter CD74^{ckO}) and CD74^{fl/fl}Foxp3^{eGFP-Cre-ERT2(-/Y)} littermates as control mice (hereafter CD74^{Ctrl}).

Reviewer Figure 4: Generation and characterization of Foxp3-specific CD74 conditional knock-out mice (CD74^{CKO}). **A:** CD74^{flox} mice were generated by the insertion of loxP sequences. PCR results illustrate the WT versus mutant alleles in heterozygous (CD74^{wt/fl}) or homozygous (CD74^{fl/fl}) mice. **B-C:** CD74^{fl/fl} mice were crossed with Foxp3^{eGFP-Cre-ERT2} mice. GFP expression was analyzed in CD4⁺CD25⁺ cells in non-fixed blood cells offspring harboring all possible genotypes for the Foxp3^{eGFP-Cre-ERT2} modification. **B:** Example of CD4⁺CD25⁺ gating strategy and GFP expression in Cre-ERT2^{-/-} or ^{-/+} or ^{+/+} by flow cytometry. **C:** Quantification of % of GFP⁺ cells in CD4⁺CD25⁺ cells according to genotype. **D-F:** Schematics: mice were treated by tamoxifen to induce the Cre-mediated CD74 deletion and spleens were analyzed by flow cytometry 4 days after the last injection. Representative dotplots (**E**) and quantification (**F**) of the % of total CD74⁺ (surface and intracellular) among CD4⁺Foxp3⁺ Treg, CD4⁺Foxp3⁻ Tconv, and CD8⁺ T cells. Statistical analyses were performed using an unpaired t-test; with pVal < **:0,001; ***:0,0001; and horizontal lines represent median.

These results (**Reviewer Figure 4**) are now included as **Figure S8** in the revised version of the manuscript.

Impact of CD74 deletion on tumor Tregs,

To analyze the impact of CD74 deletion on tumor Tregs, CD74^{Ctrl} and CD74^{CKO} mice were treated with tamoxifen (on day 0, 7 and 14), grafted with MCA or MC38 tumor cells on day 7 and tissues were analyzed on day 18 by flow cytometry (**Reviewer Figure 5A**). In both tumor types, total CD74 (surface + intracellular) was expressed in an important proportion of tumor-Tregs, and at lower levels in Tregs from other tissues (spleen, TDLN, liver) while CD4+ Tconv and CD8+T cells expressed CD74 only in the tumor (**Reviewer Figure 5B, C**). Tamoxifen administration induced a significant deletion of CD74 mainly in Tregs in all tissues, in a similar pattern for both tumor types. Although there was a tendency to a lower ratio of CD74KO vs control Tregs in the tumor compared to the spleen in the MCA model (**Reviewer Figure 5D-left**), no differences were observed for the MC38 tumors (**Reviewer Figure 5E-left**, with a very high dispersion in the tumor). However, a consistent significant drop in the level of Foxp3 expression (Geomean) was detected in the Tregs in both tissues, and for the two tumor models analyzed (**Reviewer Figure 5D-E-right**). These data reinforce the working hypothesis that CD74 is upregulated by Tregs in the tumor, and that it contributes to the maintenance of Foxp3 stability.

These results (**Reviewer Figure 5A, 5B-C and 5D-E**) are now included as **Figure 7E, S8.G-H and 7F-G**, respectively, in the revised version of the manuscript.

Altogether, results obtained in syngeneic mouse models validate our initial observation that, as in humans, CD74 plays an important role for the adaptation of Tregs to the tumor microenvironment and for the maintenance of Foxp3 stability.

Reviewer #1: In addition, there is already a considerable number of studies that investigated a large diversity of biological functions of CD74 in various cells of the immune system, including T and B cells, apart from its core role as a chaperone in HLA-II loading. Moreover, molecular binding partners, signaling circuits and transcriptional programs related to CD74 activation have been described that roughly match those characterized by the authors for Treg. While these observations were not specifically reported on Treg, they somewhat dampen the fundamental character of the findings described here. In order to provide such substantial step forward, the authors could #1) dissect the cause for the supposed different expression of CD74 in Tregs versus Tconv/CD8TC, #2) clarify whether and why CD74 mediates Treg immigration or alternatively Treg maintenance in the tumor or how FoxP3 stabilization is affected on a molecular level.

As discussed in this reviewer's comment, CD74 has been mainly studied in APCs for its role as MHC-II chaperone. Initially, CD74 was described as a key molecule involved in antigen presentation, and then as a key mediator of B cell survival and development. Later, its function as co-receptor of MIF was unveiled, broadening its role to immune-cell motility, adhesion, and mediator of intracellular pathways impacting the biology of macrophages, B and T cells. Here, we uncover a new function for CD74, starting from the observation that tumor Tregs express the highest CD74 levels compared to other T cells in the tumor and in the blood.

#1) dissect the cause for the supposed different expression of CD74 in Tregs versus Tconv/CD8TC

To address Reviewer 1's suggestions and given that in the original version of the article we had not analyzed CD8+T cells, we have now **extended our quantification analysis to human CD8+ T cells** in blood, tumor-draining lymph nodes (TDLN) and tumor tissues from paired NSCLC patient (n=4); in spleen and tumor from MDA-MB231 tumor-bearing NSG mice transferred with human T cells (n=5); and to mouse CD8+ T cells in spleen, TDLN, and tumor from B16 tumor-bearing mice (n=4) (**Reviewer Figure 6**). Across all tumor types, we consistently observed that the surface expression of CD74 protein, is most prominent on Tregs, particularly within the tumor microenvironment, compared to Tconvs and CD8+T cells.

Reviewer Figure 6A is now included as Fig.1D-E and S1E in the revised version of the manuscript, Reviewer Figure 6B is now included as Fig.S1F in the revised version of the manuscript and Reviewer Figure 6C is now included as Fig.7A-left and 7B-top in the revised version of the manuscript.

Overall, these additional analyses bolster our initial observation, providing further confirmation that, among T cells, CD74 expression at the protein level is most pronounced on Tregs, especially within the tumor microenvironment; and this observation extends to include different human tumor types and different mouse cancer models.

Additionally, seeking clarification on the potential mechanism underlying differential expression of CD74 in Tregs versus Tconvs, we addressed potential regulatory mechanisms at the chromatin level. To achieve this, we leveraged single-cell ATACseq data obtained from human CD4+ Tconvs and Tregs that were sorted from both blood and tumors of NSCLC patients (see Figure 1, data re-analyzed from Tosello et al. submitted manuscript). As shown in **Reviewer Figure 7**, Tregs and Tconvs differ in the chromatin accessibility patterns of the CD74 gene at different levels accordingly to the tissue. While only one intronic peak is more accessible in blood Tregs compared to blood Tconvs (Pval \leq 0.05 & Log2FC \geq 0.5) (chr5:150409694-150410194); four peaks are more accessible in tumTregs compared to tumTconvs (chr5:150397090-150397590, chr5:150397621-150398121, chr5:150398582-150399082, chr5:150399367-150399867), all corresponding to intronic regions of CD74. Notably, in Tregs two of these opened peaks (chr5:150398582-150399082, chr5:150399367-150399867) correlate with CD74 gene expression, and are predicted to bind several transcription factors, including notably FOXP3, and C/EBPbeta, c-Jun, c-Ets-1, GR, and STAT4 among others (Granja et al., 2021; Messeguer et al., 2002; Farré et al., 2003).

Overall, these results suggest that distinct CD74 expression in Tregs and Tconvs is partly driven by epigenetic factors.

These results (**Reviewer Figure 7**) are now included as **Fig.1F** in the revised version of the manuscript.

#2) clarify whether and why CD74 mediates Treg immigration or alternatively Treg maintenance in the tumor or how FoxP3 stabilization is affected on a molecular level.

Concerning the **effect of CD74 on Treg migration**, we are convinced that CD74 does play a role in Treg migration, as our results showed that in vitro, CD74KO Tregs have reduced motility and disturbed trajectories in both, microchannels and micropillars.

Now, to better understand how CD74 could be **impacting Treg maintenance in the tumor or FoxP3 stabilization** on a molecular level, we have performed additional experiments:

- To evaluate CD74 impact on Treg resistance to apoptosis upon activation in the presence of tumor supernatant, and/or IL-12 - a cytokine known to induce Treg destabilization (Zhuo et al., 2012; Feng et al., 2011) , we cultured WT and CD74KO Tregs in their presence and assessed Caspase-3 induction by FACS (**Reviewer Figure 8A**). We observed that activation of Tregs in the presence of tuSN alone or together with IL-12, significantly induced apoptosis in CD74KO, but not in WT Tregs, suggesting that CD74 contributes to increased Treg survival in the presence of tumor-derived soluble factors.

- Additionally, we measured the methylation level of the Treg-specific demethylated region (TSDR) of Foxp3, which is an evolutionary conserved non-coding element 2 present in the Foxp3 gene locus, which, when demethylated, sustains Foxp3 expression (Cuadrado et al., 2018) (**Reviewer Figure 8B**). We observed that upon expansion with α CD3 α CD28-coated beads and IL-2 (control condition), WT and CD74KO Tregs show similar percentages of Foxp3 TSDR demethylation (99.96% and 99.79% respectively), and that CD4+ T cells have a much lower Foxp3 TSDR demethylation level (0.62%) than Tregs. However, in the presence of tuSN, TSDR demethylation decreased more in CD74KO Tregs (53.33%) compared to WT Tregs (69.40%). This difference was deepened with the addition of IL-12, as Foxp3 TSDR was demethylated at 10.26% in CD74KO Tregs, against 47.57% in WT Tregs. This result provides a molecular basis for the CD74-mediated maintenance of Foxp3 stability in Tregs.

Taken together, our initial data and this newly generated information suggest that CD74 is involved not only in Treg migration but also in the maintenance of Tregs in the tumor. Furthermore, this latter effect is partly mediated by protecting Tregs from apoptosis and by sustaining FoxP3 stability.

These results (**Reviewer Figure 8A and 8B**) are now included as **Fig 4F** and **Fig4H**, respectively, in the revised version of the manuscript.

Reviewer #1: Here, the authors fail to embark on previous studies demonstrating e.g. NF κ b activation through CD74 signalling or exploring the role of MIF or other natural CD74 ligands and their abundance in the tumor microenvironment. Still, the reported findings are of particular translational interest and will attract broader attention of researchers in the field of cancer immunology and immunotherapy.

An additional role of the membrane CD74 is provided by its function as MIF co-receptor when associated with CD44 or CXCR4 (Schwartz et al., 2009; Shi et al. 2006). Thus, we quantified the co-expression of CD44 and CXCR4 with CD74 on the surface of T cells in vivo (**Reviewer Figure 9**). For this purpose, NSG mice were engrafted with 5×10^6 MDA-MB231 cells, and 14 days later, PBMCs were injected intravenously along with in vitro expanded Tregs (expTregs). Six days later, the spleen, liver, and tumor were analyzed by FACS. We observed that all Tregs express CD44 on their surface in the spleen, liver, and tumor, and that tumTregs express an increased level of surface CXCR4 compared to splenic and hepatic Tregs, resembling the CD74 surface expression pattern. In more details, the proportion of double-positive CD74+CD44+ and CD74+CXCR4+ is elevated in tumTregs (19.43+/-5.35% and 13.94+/-8.88%, respectively) and hepatic Tregs CD74 (20.00+/-9.25% and 13.13+/-9.19%, respectively), compared to splenic Tregs (5.62+/-2.40% and 1.15+/-1.15%, respectively). **These results suggest that in liver and tumor, upregulation of CD74 on the surface of Tregs could be associated with its role as a co-receptor for MIF** (Lue et al., 2010; Jankauskas et al., 2019).

These results (**Reviewer Figure 9A and B**) are now included as **Fig.S5G and S5H**, respectively, in the revised version of the manuscript.

- Additionally, following the reviewer request to evaluate MIF-induced NF- κ B activation in Tregs, we cultured WT and CD74KO Tregs with soluble MIF (100ng/mL) (Starlets et al., 2006) or IL-1b (positive control) and assessed NF- κ B and AKT phosphorylation over time by western blot (**Reviewer Figure 10**). No consistent activation of these transcription factors was observed in the two analyzed donors. These data suggest a limited or context-dependent impact of MIF on NF- κ B and AKT activation in Tregs, necessitating further exploration. We include these results for reviewer perusal but would not favor to include them in the modified manuscript.

- Furthermore, to investigate the potential impact of MIF ligation on CD74 on Treg stability, we incubated WT or CD74KO Tregs with soluble MIF, with or without MDA-MB231-derived supernatant. Following a 48-hour incubation period, we assessed Foxp3 levels, but no discernible differences were observed (**Reviewer Figure 11A**), indicating that under the conditions tested, MIF ligation on CD74 may not play a significantly role in modulating Foxp3 expression and, consequently, Treg stability.

- Finally, to evaluate MIF effects in vivo, we generated MDA-MB231 cell lines KO for MIF expression using a CRISPR-Cas9 RNP targeting MIF at exon 5 2, and the corresponding control MDA-MB231 cells using a control RNP. We verified the efficacy of MIF targeting by measuring MIF concentration by ELISA in the supernatant of the tumor cells 4 days after MIF deletion, and observed that although MIF concentration was significantly reduced, it was not completely abrogated at this time point (**Reviewer Figure 11B**). Control and MIFKO tumor cells were injected in NSG mice, and 10 days later we co-transferred in the same mouse a mix of PBMCs, WT and CD74KO Tregs (**Reviewer Figure 11C**). We observed that both tumors developed similarly in NSG mice, indicating that MIF KO did not affect per se the tumor growth capacity (**Reviewer Figure 11D**). Seven days after transfer, we quantified tumor Tregs. As in the previous experiments, higher amounts of WT Tregs were observed compared to CD74KO Tregs in both WT and MIFKO tumors with no major differences between the two tumors (**Reviewer Figure 11E-G**). While the remaining MIF production by MIFKO tumor cells might mask a potential impact of MIF on Treg recruitment, these findings do not suggest a major role for CD74 in the response of Tregs to MIF.

Overall, these series of experiments do not support a significant role for MIF ligation on CD74 in contributing to Treg stability in the tumor.

Reviewer Figure 11: Evaluation of MIF effect on Tregs. A: Quantification of Foxp3 geometric mean in WT and CD74KO Tregs after stimulation with MDAMB231-tumor cell derived supernatant (tuSN) with or without soluble MIF (48h, 200ng/mL). **B:** MDAMB231 cells were electroporated with a CRISPR-Cas9 RNP targeting MIF at exon 2 (MIFKO, orange) or with a control RNP (ctrl, black). Concentration of MIF was evaluated by ELISA in the MDAMB231-derived supernatant 4 days after CRISPRCas9 deletion of MIF. **C-G:** MIFKO or ctrl MDAMB231 cells were engrafted in NSG mice, and 10 days later, PBMCs, WT expTregs and CD74KO expTregs are injected intravenously. Six days later, spleens, livers and tumors were analyzed by FACS. **D:** Measurement of tumor size of MIFKO or ctrl tumor in NSG mice

transferred with T cells. Representative dotplots (E), quantification of the absolute number of WT expTregs (CD74+, blue) and CD74KO (CD74-, red) cells among total expTregs in the MIFKO tumor (orange) or ctrl tumor (black) (F), and ratio of CD74KO/WT Tregs infiltrating the tissues (G). Statistical analyses are performed using an unpaired t-test with pVal < **:0,001; ***:0,0001

These results (Reviewer Figure 11A and B-G) are now included as Fig.S5I and J-N, respectively in the revised version of the manuscript.

Reviewer #2 (Remarks to the Author): with expertise in CD74

The paper by E. Bonnin et al. reveals yet another function of the CD74 molecule in a very special cell type, the regulatory T-cells.

They show in a xenograft mouse model that human Treg that reside in human breast carcinoma transplants inhibit the GVHD-induced tumor rejection brought about by coinoculating these mice with human PBMCs.

They further show by a CRISPER-Cas9 knock out of CD74 in human Treg that the inhibitory effect of these manipulated cells on GVHD-induced breast carcinoma rejection is impeded, allowing the conclusion that CD74 is causative. This assumption is tested by forced hyperexpression of CD74 in Treg and the result supports this concept in that these CD74 hyperexpressing cells exert an even stronger inhibitory effect on tumor rejection. They proceed in explaining this effect by a change in migratory capacity of these CD74KO Treg which they show by ex vivo migration assays are unable to properly migrate. The last set of experiments focusses the intracellular changes in Treg when CD74 is knocked down. Forced CD74 deficiency leads to structural alterations, a hairy phenotype, cytoskeletal changes and altered gene expression in a number of genes.

The paper is perfectly written and the data as they are presented in a large number of complex Figures are convincing as such.

However, I am especially surprised that the different phenotypes of isolated and genetically manipulated Treg seem to be stable over time ex vivo and that these phenotypes seem to be rather traits than different states of activation since these isolated and manipulated sub-sets of Treg can even be expanded by repetitive IL-2 treatment. This further implies that activation induced death does not occur.

In my opinion, this complex study is a high-wire performance in highly artificial systems. Looking at it from far below as a grounded pathologist who performs PD-1 immunohistochemistry every day to detect and quantify intratumoral T-cells expressing PD-1, this study lacks the connection to the ground, i.e., to the real life situation in cancer patients. Most carcinomas and melanomas contain very few tumor infiltrating T-cells and even less Treg.

We appreciate this reviewer's thorough evaluation of our manuscript and comments, and we are pleased that this reviewer finds our manuscript well-written with convincing data.

We acknowledge this reviewer's concern regarding the stability of different phenotypes in isolated and genetically manipulated Treg over time ex vivo. The observed stability, and the absence of activation-induced death, is indeed an interesting aspect of our findings. Our interpretation is that CD74 expression is pivotal to Treg biology, including survival, stability, and suppressive function, particularly in the challenging conditions of the tumor

microenvironment (TME) characterized by factors like hypoxia, low nutrient levels, and inflammatory molecules. In this context, the absence of CD74 appears indispensable for Treg adaptation and function. Conversely, in *in vitro* settings, where conditions have been optimized to facilitate Treg expansion and sustained survival—supported by a continuous supply of exogenous IL-2—reveal a less pronounced role for CD74 in Treg homeostasis. This nuanced distinction emphasizes the contextual importance of CD74 in Treg dynamics, dependent upon the environmental challenges encountered.

We understand this reviewer's concern about the potential gap between our experimental setup and the clinical reality of cancer patients. Nevertheless, our discovery of CD74's novel role in tumor Tregs, brings new knowledge on a cell population that, although present in relatively low numbers in the TME, plays a critical role in tumor development and progression, and have a high therapeutic potential. We hope that the novel experiments included in this revised version of the manuscript, including data in immunocompetent mice and a description of underlying molecular mechanisms, will convince this reviewer on the relevance of this work to the scientific community.

Reviewer #3 (Remarks to the Author): with expertise in Tregs, cancer immunology

The authors demonstrate that Tregs express surface CD74 in a tumor microenvironment-dependent manner. CD74 specifically stabilizes the suppressive phenotype of Tregs within tumors and is required for the migration and retention of intratumoral Tregs. The authors provide evidence that Treg retention in tumor may reflect CD74 triggered reorganization of Tregs' intracellular organelles and capacity to interact with neighboring cells. As the invariant chain of MHC-II, CD74 was initially thought to be mainly restricted to antigen presenting cells (APCs). The authors report for the first time that CD74 is exclusively expressed by tumor infiltrated Tregs and specifically stabilizes their suppressive phenotype. These findings represent a potentially important conceptual advance in our understanding of Treg biology in the tumor microenvironment and also provide insight into developing therapeutic approaches targeting CD74. However, there are critical issues that need to be addressed before consideration for publication.

Major critique:

Reviewer #3 1. The authors claim that CD74 specifically regulates the Treg suppressive phenotype in tumor according to the observation that CD74KO Tregs are exclusively dysfunctional in the tumor tissue, but not *in vivo* in a GVHD model. However, the suppressive activity of CD74KO Tregs in GVHD may not be an appropriate model for comparison, since the allo-reactive response in the GVHD model lacks or masks the tumor cues that may upregulate Treg surface CD74 expression. A direct comparison between intratumoral Treg and in non-tumor tissues from Treg CD74 expression and the Treg phenotype would be more informative.

In response to the reviewer's comment, we performed a direct comparison of the phenotype of CD74KO expTregs in the spleens, livers and tumors of MDA-MB231- bearing NSG mice transferred with PBMCs and WT or CD74KO expTregs, six days after T cell transfer. As shown in **Reviewer Figure 12**, we observed that: i) the proportion of liver Tregs expressing CD74 at

the surface was elevated compared to splenic Tregs, yet lower than in tumor Tregs (**Reviewer Figure 12A**); and ii) as previously described for the spleen, CD74 deletion did not induce significant changes in CD25 or 4-1BB expression in Tregs from the liver, but significantly affected Tregs from the tumor (**Reviewer Figure 12B**).

The results showing that CD74 expression is higher in liver Tregs than in spleen Tregs, suggest that CD74 upregulation could underly an adaptive response of Tregs to residency in non-lymphoid tissue. Nevertheless, even with this increase, liver Tregs do not exhibit a clear dysfunctional phenotype (conserved CD25 and 4-1BB expression) compared to tumor tissue Tregs. This highlights a specific role for CD74 in Treg biology within the tumor environment, different from its role in non-tumor tissues.

These results (**Reviewer Figure 12 A and B**) are now included as **Fig.S4G and S4H**, respectively, in the revised version of the manuscript.

Reviewer #3. 2. A portion (6-22%) of intratumoral Treg express CD74, while 80%-99% of conventional T cells and Tregs express intracellular CD74. Is there a functional difference between surface CD74 and intracellular CD74? It is critical to distinguish the function of different forms of CD74 in intratumoral Treg using anti CD74 that blocks surface CD74 and spares intracellular forms.

This will also rule out the potential impact of intracellular CD74.

To address the question of whether there is a functional difference between surface CD74 and intracellular CD74, we used an anti-CD74 antibody (Milatuzumab) that blocks surface CD74 to assess the impact of CD74 surface expression on tumTregs infiltration in our tumor

in vivo model. Milatuzumab, is a humanized IgG1k monoclonal antibody antagonist of CD74, that demonstrated anti-proliferative and apoptotic effects in B-cell lymphoma by blocking CD74 signaling on B cells (Frölich et al., 2012; Martin et al., 2015) (MDA-MB231 tumor-bearing mice co-transferred with PBMCs and WT expTregs were treated with three rounds of anti-CD74 antibody (Milatuzumab) administered either intravenously (i.v.) or intratumorally (i.t.). **Reviewer Figure 13** illustrates that both i.v. and i.t. administration of anti-CD74 resulted in a significant reduction of expTregs in the tumor compared to the untreated mice. Conversely, a non-significant decrease in CD4+ T cells was observed, in accordance with the high and low levels of CD74 surface expression on these cells.

These findings suggest that surface CD74 plays a critical role in tumTregs infiltration, supporting the notion that targeting surface CD74 with Milatuzumab has a specific impact on intratumoral Tregs.

These results (**Reviewer Figure13 A and B**) are now included as **Fig.S5B and S5C**, respectively, in the revised version of the manuscript.

Reviewer #3. 3. “The enhanced tumor rejection observed with the transfer of CD74KO expTregs suggests that in the absence of CD74, Tregs no longer suppress antitumor immune responses but may further acquire anti-tumoral functions.” Does this mean KO CD74 converts Treg into effector-like T cells? Since CD74 deletion does not affect in vitro or in vivo expression of key molecules, the authors should characterize the phenotype of CD74 KO Treg in this model in terms of acquisition, e.g., acquisition of effector cytokine expression (e.g., IFN γ) or reduction of suppressive cytokines (e.g., IL-10, TGF-beta).

The inquiry regarding whether CD74 deletion transforms Tregs into effector-like T cells was initially addressed in the manuscript by examining the impact of CD74 genetic loss on FoxP3 expression, based on the observation substantiated by relevant literature that a reduction in Foxp3 levels is indicative of Treg destabilization and the potential acquisition of effector function (Overacre-Delgoffe et al., 2017; Feng et al., 2011). Of note, we consistently observed a recurrent decrease in Foxp3 levels in CD74KO Tregs, both ex-vivo from tumors and in vitro upon stimulation with IL-12 or tumor-derived supernatants.

In response to the reviewer's suggestion, we conducted additional experiments to further characterize the phenotype of CD74KO Tregs. Specifically, we expanded WT and CD74KO Tregs from three HD PBMCs, re-stimulated them in the presence of MDA-MB231 derived supernatant (tuSN) alone, or together with IL-12; and assessed IFN γ production. **Reviewer figure 14** shows the proportion of IFN γ + cells. As expected, incubation with tuSN +/- IL-12 induced higher IFN γ production, with the maximal increase observed in CD74KO Tregs (**Reviewer Figure 14A**). Furthermore, this elevation in IFN γ production correlated with a decrease in Foxp3 geometric mean (**Reviewer Figure 14B**).

Collectively, these findings suggest that CD74 deletion in Tregs may indeed lead to a shift towards an effector-like phenotype, characterized by enhanced IFN γ production. These findings support the notion that CD74 may play a crucial role in maintaining Treg identity and suppressive function within the tumor microenvironment.

Reviewer Figure 14: Deletion of CD74 promotes the acquisition of a Teff-like phenotype by Tregs under destabilizing conditions. CD74KO (red) and WT (blue) Tregs from HD PBMCs (n=3 donors) were expanded with α CD3 α CD28-coated beads and IL-2 (1000IU/mL) (control). At day 7, cells were re-stimulated with α CD3 α CD28-coated beads and IL-2 (1000IU/mL), and at day 10, MDA-MB231 derived supernatant (tuSN) was added alone, or together with IL-12 (20ng/mL). At day 11, cells were incubated with Brefeldin-A for 4 hours and analyzed by FACS. Representative FACS dotplots (left panels) and quantification (right panels) of the percentage of IFN γ (A), and geometric mean of Foxp3 (B) in WT and CD74KO Tregs. Statistical analyses were performed using paired t-test with pVal < *:0,01; **:0,001; ***:0,0001; ****:0,00001.

These results (**Reviewer Figure 14A and B**) are now included as **Fig 4F and G**, respectively, in the revised version of the manuscript.

4. The authors indicate that CD74 may act as a receptor for chemokines produced by the tumor, which accounts for the retention of Tregs within tumors. Perhaps the authors could discuss potential mechanisms, e.g, chemokine-CD74 interaction that might underlie Treg migration.

This point has also been raised by reviewer #1.

We have carefully addressed this question, specifically the role of MIF, but have not found a major implication of this factor in Tregs' migration into the tumor.

Please refer to answers to reviewer#1, pages 13-15, figures 9, 10, and 11 of this document.

5. Bulk RNA-seq analysis profiling WT and CD74KO Tregs was used to define the molecular mechanisms underlying the effect of CD74 on Treg motility and tumor retention. The authors should validate DEG pathway gene products at the protein level to validate these conclusions. Following the reviewer comments, we searched for available commercial Abs that could be used by FACS to validate the candidate proteins, and found 3 Abs directed against FOXC2, LGR5 and pan Cytokeratin. We used WT- and CD74-KO Tregs cultured in-vitro with anti-CD3/anti-CD28 beads and 500UI/mL of IL-2 for the analysis (**Reviewer Figure 15**). In contrast with HLA-DR expression (used as control), for the 3 new targets we observed a higher expression of the protein in CD74KO Tregs compared to WT Tregs, as inferred from the DEGs list. Although shift in the MFI can be observed, especially for FOXC2, we believe that the quality of these staining is not appropriate for publication. We encountered difficulties in finding antibodies with validated positive staining, and there is uncertainty regarding their specificity. To properly validate them, it would be necessary to generate cell lines that both express and not these targets, but this process would require a significant amount of time. Thus, we decided to show the results for the reviewer perusal, but not to include these data in the revised manuscript.

6. For the co-culture experiment in Fig.S7, it's not clear how Treg contacting neighboring cells by exchange mitochondria contributes to Treg stability and retention in the tumor. Although

mesenchymal stem cells may transfer mitochondria to Treg to enhance their immunosuppressive function, it may not apply to the PBMC-Treg contacting. Moreover, the co-culture assay was performed in a tumor-free setting, which cannot mirror the environmental cues in the tumors.

In response to the reviewer's comment regarding the co-culture experiment depicted in Fig. S7G, we acknowledge the valid concerns raised. Upon careful consideration, we have decided to remove this result from the paper. We recognize that the observed exchange of mitochondria between PBMCs and Tregs may not be directly applicable to the tumor microenvironment. We value the reviewer's perspective, and the removal of this result aligns with the need for a more robust experimental context to better reflect the environmental cues within tumors.

Minor points:

1. Introduction: Reference #1 is a submitted manuscript, which is not appropriate as cited literature. The authors should state that the relevant findings are in a manuscript currently under consideration.

Done

2. Fig. S5B-D: A CD74KO group should also be included in addition to CD74OE group. It will be more convincing to compare the impact of CD74KO and CD74OE on Treg tumor accumulation side-by-side.

We appreciate the suggestion to include a CD74KO group in addition to the CD74OE group. However, it's important to note that the CD74OE and CD74KO cells are generated through different methods, specifically lentiviral transduction for CD74OE and CRISPR/Cas9 for CD74KO. The distinct approaches for overexpression and knockout make a direct side-by-side comparison challenging and potentially confounding. Consequently, we believe that the comparison between CD74OE and CD74KO groups may not be directly applicable or provide meaningful insights into the impact of CD74 modulation on Treg tumor accumulation.

Nota bene: bulk-RNAseq for CD74KO and WT Tregs has been deposited in GEO database with the GEO accession number: GSE229389. For the reviewers access, the token is "knmnamkcrvmhpsz."

REVIEWERS' COMMENTS

Reviewer #1 (Remarks to the Author):

The authors have conducted a substantial body of adequate in vivo experiments to address my concerns. The new results in principle support their major conclusions, although the overall effect of CD74 on Treg accumulation in syngeneic murine tumors appears weak (particularly in inducible CD47KO Treg) which suggests that CD74 acts more on FoxP3 stabilization than on Treg migration or maintenance.

I still have difficulties in interpreting new figure 7C and D, relating to the key experiment with transfer of CD47KO Treg into tumor bearing mice: While the dotplot 7A is meant to show a representative distribution of CD74KO vs WT Treg in a tumor, it demonstrates even more CD74 KO Tregs than WT Tregs, although injected at a 1:1 ratio. This seems not reflected in the cumulative results shown in 7D, where all 3 animals show increased proportions of WT Tregs in the tumors. The authors should clarify this seeming contradiction.

Reviewer #3 (Remarks to the Author):

The authors' response to our comments is appropriate and satisfactory.

To follow on the comment of reviewer 1,

Reviewer #1 (Remarks to the Author):

The authors have conducted a substantial body of adequate in vivo experiments to address my concerns. The new results in principle support their major conclusions, although the overall effect of CD74 on Treg accumulation in syngeneic murine tumors appears weak (particularly in inducible CD47KO Treg) which suggests that CD74 acts more on FoxP3 stabilization than on Treg migration or maintenance.

I still have difficulties in interpreting new figure 7C and D, relating to the key experiment with transfer of CD47KO Treg into tumor bearing mice: While the dotplot 7A is meant to show a representative distribution of CD74KO vs WT Treg in a tumor, it demonstrates even more CD74 KO Tregs than WT Tregs, although injected at a 1:1 ratio. This seems not reflected in the cumulative results shown in 7D, where all 3 animals show increased proportions of WT Tregs in the tumors. The authors should clarify this seeming contradiction

We understand the reviewer's comment, as the graph in Fig. 7C/D is not self-explanatory. The problem stems from the fact that the initial mix of WT/CD74KO Tregs injected into the mice was not 50 :50 as originally planned. Although we counted the cells before mixing them, in these particular experiments, the final injected mix was of 84/16 for replicate-1 (n=1 mouse) and of 71/29 for replicate-2 (n=2 mice). Consequently, to take into consideration the variability of the CD45.2/CD45.1 ratio in the injected mix, results were expressed as the CD45.2/CD45.1 ratio in the tissue, normalized to the initial CD45.2/CD45.1 ratio (as depicted in Fig. 7D).

Thus, so as not to confuse the reader in Figure 7C, we propose to remove the exemplar dot plots for the spleen and tumor of one mouse, and to precise the initial ratio in the mix used for each replicate in the Method section (see below), and changed the figure legend accordingly.

For the reviewer perusal, we show here the representative dot plots of the distribution of CD45.2 CD74KO/CD45.1 WT Tregs for the replicate 1 contained in the initial injected mix in addition to the distribution obtained in the spleen and tumor tissues, previously shown in the original figure7C.

Revised Method section:

Adoptive transfer of CD4+CD25+ cells in syngeneic tumor model

Rag2^{-/-} KP mice (43) are subcutaneously injected on the flank with 100μL of PBS containing 0.5x10⁶ MCA101 tumor cells. On day 12, mice receive a mix of CD45.2+ CD74KO CD4+CD25+ splenocytes (obtained from CD74 full KO mice) and CD45.1+ CD4+CD25+ splenocytes (obtained from WT littermates). CD4+CD25+ splenocytes were obtained by successive enrichment using magnetic columns (negative selection of CD4+ T cells (MojoSort, Biolegend #480005) followed by CD25 positive selection (MicroBead, Miltenyi #130-091-072), and a mix of 1x10⁶ cells are injected in 100μL by i.v. On day 18 (6 days after cell transfer), spleens and tumors are collected for cytometry analysis. The injected mix of CD45.2 CD74KO/CD45.1 WT CD4+CD25+ splenocytes contained a proportion of 84/16 for replicate-1 (n=1 mouse) and of 71/29 for replicate-2 (n=2 mice). To take into consideration the variability of the CD45.2/CD45.1 ratio in the injected mix, results are expressed as the CD45.2/CD45.1 ratio in the tissue, normalized to the initial CD45.2/CD45.1 ratio.